# Discourse-Aware Retrieval-Augmented Generation via Rhetorical Structure Modeling

## Abstract

Retrieval-Augmented Generation (RAG) has emerged as an important means for enhancing the performance of large language models (LLMs) in knowledge-intensive tasks. However, most existing RAG strategies treat retrieved passages as flat and unstructured text, which prevents the model from capturing structural cues and constrains its ability to synthesize dispersed evidence and to reason across documents. Although a few recent approaches attempt to incorporate structural signals, each remains restricted to shallow representations such as entity graphs or dependency edges and thus fails to capture hierarchical discourse organization. To overcome these limitations, we propose Discourse-RAG, a structure-aware framework that explicitly injects discourse signals into the generation process. Our method constructs intra-chunk rhetorical structure theory (RST) trees to capture local coherence hierarchies and builds inter-chunk rhetorical graphs to model cross-passage discourse flow. These structures are jointly integrated into a planning blueprint that conditions the generation. Experiments on question answering and long-document summarization benchmarks show the efficacy of our approach. Discourse-RAG achieves a new state-of-the-art ROUGE-L score of 42.4 on ASQA dataset and improves LLM Score by 12.79 points over standard RAG on Loong benchmark. These findings underscore the important role of discourse structure in advancing retrieval-augmented generation. Code is available at https://anonymous.4open.science/r/Discourse-RAG.

## 1 Introduction

The advent of large language models (LLMs), including LLaMA (Touvron et al., 2023), Qwen (Yang et al., 2025), and GPT series (Achiam et al., 2023), has promoted research progress in Natural Language Processing (NLP), achieving competitive performance across a wide range of tasks such as question answering (Wu et al., 2025a; Lee et al., 2025a; Zhang et al., 2025b), summarization Mondshine et al. (2025); Liu et al. (2025a); Wang et al. (2025a); Luo et al. (2025), and text generation (Duong et al., 2025; Bigelow et al., 2025; Que & Rong, 2025; Zhang et al., 2025a). However, due to the reliance on static training corpora, LLMs are insufficient in knowledge-intensive scenarios (Chang et al., 2025; Lee et al., 2025b; Yue et al., 2025). Challenges arise in handling domain-specific knowledge, proprietary data, or information that requires real-time updates (Wang et al., 2024b; Xia et al., 2025). Retrieval-Augmented Generation (RAG) has been proposed as a suitable solution by integrating an external knowledge injection component through retrieval-based mechanisms (Lewis et al., 2020; Asai et al., 2024; Chan et al., 2024).

In terms of RAG pipelines, external documents are segmented into chunks, which are then encoded into vectors and stored in a database. At query time, relevant chunks are retrieved to provide contextual grounding for the LLM (Lewis et al., 2020). One important but insufficiently addressed limitation of existing RAG systems concerns the *mismatch between retrieval granularity and generative understanding*. While retrieval modules return semantically relevant chunks, these chunks are often fragmented in discourse, which is like

scattered pieces of evidence without clear logical connections (Edge et al., 2024; Su et al., 2025). This issue manifests at two levels. First, ***intra-chunk structural blindness***: within each chunk, models often fail to capture internal coherence. As depicted in Figure 1 (left), Chunk A mentions a "12% lower incidence," while Chunk B notes "no significant overall effect." Without recognizing that the former is a conditional finding (*e.g.,* among deficient adults in winter), the model tends to overgeneralize and incorrectly conclude that "vitamin D reduces flu risk." Second, ***inter-chunk coherence gaps***: across multiple chunks, RAG systems struggle to identify rhetorical connections between segments. This deficiency prevents effective resolution of conflicting claims, as standard approaches lack the capacity to organize retrieved evidence through higher-level discourse relations, as shown in Figure 1. Prior relevant methods, including semantic chunking (Wang et al., 2025c; Qu et al., 2025; Zhao et al., 2025) and graph-based RAG (Edge et al., 2024; Nigatu et al., 2025; Hu et al., 2025; Wu et al., 2025b; Zhu et al., 2025), aim to improve semantic connectivity (*e.g.,* linking entities) but they largely overlook the rhetorical structure that governs arguments flow, evidence presentation, and conclusions formulation. This leaves the generator to grapple with a *bag of facts* rather than a coherent *line of reasoning*.

Recent investigations have revealed that integrating discourse knowledge into LLMs can improve downstream performance (Nair et al., 2023; Gautam et al., 2024; Liu & Demberg, 2024). These findings suggest the drawback of relying solely on flat sequential representations and underline the benefits of deeper discourse modeling (Ma et al., 2025). Building on these insights, the present work investigates ***whether explicitly modeling and providing discourse knowledge to the LLM can further improve generation quality in the context of RAG***. To answer this, we suggest `Discourse-RAG`, a framework that constructs local rhetorical trees for each retrieved chunk and infers inter-chunk rhetorical relations across chunks to form a global discourse graph. To synthesize information, rather than merely concatenating it, a model needs not only to understand the relations between evidence but also to strategize how to present them. This requires a high-level plan to orchestrate the argumentative flow. Therefore, we introduce a discourse-aware planning module that enables the model to *dynamically generate* a rhetorical plan to guide the generation process. As shown in Figure 1 (right), the structure-aware process enables the model to infer that "vitamin D is

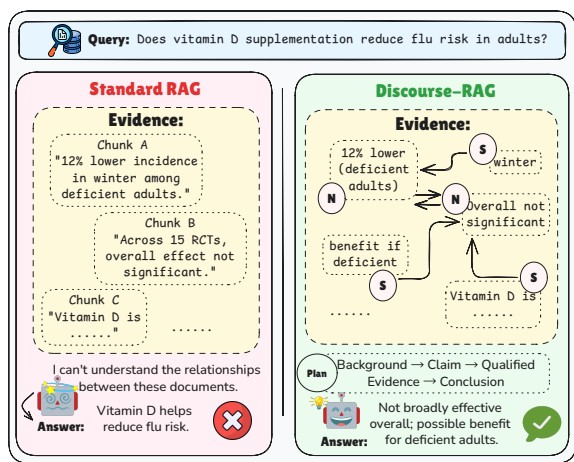

Figure 1: Comparison between standard RAG and our Discourse-RAG. While standard RAG retrieves isolated chunks without structural links, Discourse-RAG organizes evidence into rhetorical relations and plans, yielding qualified and contextually accurate answers. Here, S denotes *Satellite* (the supplementary part), and N denotes *Nucleus* (the core part).

not broadly effective but may benefit deficient adults under specific conditions", producing more faithful answers and aligned with the underlying evidence.

In our experiments, we evaluate `Discourse-RAG` on three benchmarks, `Loong`, `ASQA`, and `SciNews`. Consistent improvements are observed when compared with standard RAG systems and previously reported state-of-the-art (SOTA) methods. On the Loong benchmark, our approach delivers gains of up to +16.0 points in LLM Score under long-document settings. On the ASQA dataset, the method exceeds the best existing systems on ROUGE-L (42.4 vs. 42.0) and improves exact match and DR Score by notable margins. On the SciNews benchmark, `Discourse-RAG` establishes new SOTA performance across all evaluation metrics. In addition, our framework is training-free, which allows plug-and-play applicability across different models and tasks.

**In summary, this work offers the following contributions:**

- We present `Discourse-RAG`, a framework that explicitly injects discourse knowledge into RAG systems to alleviate the mismatch between retrieval granularity and generative understanding.
- We propose a unified structural modeling approach that combines intra-chunk RST trees, inter-chunk rhetorical graphs, and discourse-driven planning to capture local hierarchies, cross-passage coherence, and global argumentative flow.
- We conduct extensive experiments on knowledge-intensive QA and summarization tasks, demonstrating consistent gains over strong RAG baselines. Analysis studies further confirm the efficacy of discourse-aware guidance in enhancing answer correctness, coherence, and factuality.

## 2 RELATED WORK

### 2.1 STRUCTURE-AWARE RETRIEVAL-AUGMENTED GENERATION

Retrieval-Augmented Generation (RAG) enhances LLMs in knowledge-intensive tasks by retrieving external evidence (Lewis et al., 2020). However, conventional RAG methods typically treat retrieved chunks as isolated and flat sequences, overlooking their structural interconnections. To mitigate this, recent research has explored structure-aware variants of RAG. Graph-based methods such as GraphRAG (Edge et al., 2024) and KG-RAG (Sanmartin, 2024) organize evidence into knowledge graphs, while subsequent work improves retrieval by simulating human memory mechanisms (Gutierrez et al., 2024; Gutiérrez et al., 2025) or enriching graph semantics (Liang et al., 2025). Other approaches construct structured subgraphs for coherence (Mavromatis & Karypis, 2025; Li et al., 2025a), or employ alternative formats like hierarchical graphs (Zhang et al., 2024; Wang et al., 2025b; Huang et al., 2025), trees (Fatehkia et al., 2024; Sarthi et al., 2024), and tables (Lin et al., 2025). More adaptive strategies dynamically select structures based on context (Li et al., 2025b). Despite these advances, most efforts emphasize surface-level associations while neglecting rhetorical or argumentative structure. This hinders logical depth and discourse coherence, which our work seeks to address.

### 2.2 RHETORICAL STRUCTURE THEORY FOR TEXT GENERATION

Rhetorical Structure Theory (RST; Mann & Thompson (1987; 1988)) is a discourse framework that models hierarchical dependencies and rhetorical relations among Elementary Discourse Units (EDUs). It distinguishes between *nucleus* and *satellite* units, connected by relations such as *Elaboration*, *Causality*, and *Contrast*, forming tree structures that reflect communicative intent. Foundational work (Marcu, 1997; 1999; Mann & Thompson, 1987; Bhatia et al., 2015; Hayashi et al., 2016) has established strong correlations between rhetorical structure and human text planning (Adewoyin et al., 2022). Later studies leveraged RST by converting trees into dependency graphs or imposing structural constraints to improve coherence and consistency in neural generation models (Chistova, 2023; Zeldes et al., 2025; Chistova, 2024; Maekawa et al., 2024). More recent efforts have integrated RST into LLMs to improve cross-sentence reasoning and enhance both structural integrity and interpretability of generated outputs (Liu et al., 2023; Liu & Demberg, 2024). However, most prior work largely depends on task-specific fine-tuning. The present work extends RST modeling to the RAG setting by explicitly encoding the discourse structure of retrieved passages and integrating it into the generation process.

## 3 METHODOLOGY

**Task Formulating.** We formalize the Retrieval-Augmented Generation (RAG) as conditional generation. Given a query $q$ and a set of top-$k$ retrieved chunks $\mathcal{C}(q; \mathcal{D}) = c_1, c_2, \ldots, c_k$ from corpus $\mathcal{D}$, the output is:

$$y = \arg\max_{y'} P(y' \mid q, \mathcal{C}(q; \mathcal{D})), \tag{1}$$

where $P(\cdot)$ denotes the conditional distribution of the generator. To overcome the limitations of the retrieval-and-concatenation paradigm (standard RAG), which treats retrieved chunks as a flat sequence, we propose `Discourse-RAG` that augments RAG with rhetorical parsing and discourse-level planning.

As illustrated in Figure 2, our pipeline consists of three main stages. (1) we delve into each chunk $c_i$ to uncover its internal logical hierarchy by constructing an intra-chunk RST tree $t_i$, (2) we zoom out to map the relational landscape across all chunks $\mathcal{C}$ via an inter-chunk rhetorical graph $\mathcal{G}$, and (3) we apply a rhetorically-driven planning module that devises a blueprint $\mathcal{B}$ based on $\mathcal{T} = t_{i=1}^k$ and $\mathcal{G}$ to guide the final generation.

We hypothesize that *under identical retriever and decoding conditions, explicitly injecting rhetorical structures and planning improves correctness, coherence, and factual consistency*. Here, rhetorical modeling serves as a *knowledge-level prior*, while planning offers *reasoning-level guidance*, jointly inducing stronger structural biases than standard RAG. The following paragraphs provide a detailed account of each component.

**Intra-chunk RST Tree Construction.** For each retrieved chunk $c_i$, we construct an RST tree $t_i$ using an LLM-based RST agent $\mathcal{A}$ to model the local coherence.[1] Given $c_i$, the agent jointly performs elementary discourse units (EDUs) segmentation and rhetorical parsing, producing: (1) a sequence of EDUs $\{e_{i_1}, \ldots, e_{i_m}\}$, (2) nucleus and satellite roles assignments, and (3) rhetorical relations among EDUs. Formally:

$$c_i \xrightarrow{\mathcal{A}} \{e_{i_1}, e_{i_2}, \ldots, e_{i_m}\}, \quad t_i = (V_i, E_i), \tag{2}$$

where $V_i = \{e_{i_1}, \ldots, e_{i_m}\}$ is the set of EDU nodes, $\mathcal{R}$ is the set of rhetorical relations (*e.g., Elaboration, Contrast,* and *Cause*), and $E_i \subseteq V_i \times V_i \times \mathcal{R}$ is the set of directed connections labeled with relation types. The symbol $\times$ denotes the *cartesian product*. Figure 2 illustrates how EDUs are organized into a hierarchical tree. The parsing process is formalized as a conditional generation problem:

$$P(t_i \mid c_i; \theta_{\mathcal{A}}) = \prod_{j=1}^{m} P(e_{i_j} \mid c_i; \theta_{\mathcal{A}}) \cdot \prod_{(u,v)} P(r_{u,v} \mid e_{i_u}, e_{i_v}, c_i; \theta_{\mathcal{A}}), \tag{3}$$

where $P(e_{i_j} \mid c_i)$ signifies the probability of EDU boundary prediction and $u, v \in V_i = \{e_{i_1}, \ldots, e_{i_m}\}$ are discourse units, $P(r_{u,v} \mid e_{i_u}, e_{i_v}, c_i)$ corresponds to the probability of the rhetorical relation between two EDUs, and $\theta_{\mathcal{A}}$ denotes the parameters of the LLM agent.

**Inter-chunk Rhetorical Graph.** We construct a directed graph $\mathcal{G} = (\mathcal{C}, \mathcal{F})$, where $\mathcal{C}$ represents the node set, each representing a retrieved chunk $c_i$. Edges set $\mathcal{F} \subseteq \mathcal{C} \times \mathcal{C} \times (\mathcal{R} \cup \text{Unrelated})$ denote rhetorical relations or lack thereof. These inter-chunk connections are inferred via an LLM-based agent $\mathcal{A}$,[2] which performs pairwise comparison and assigns a discourse label $r_{i,j}$ or marks the pair as `Unrelated`:

$$c_i, c_j \xrightarrow{\mathcal{A}} r_{i,j}, \quad r_{i,j} \in \mathcal{R} \cup \{\text{Unrelated}\}. \tag{4}$$

The complete graph construction is formalized as a probabilistic modeling task:

$$P(\mathcal{G} \mid \mathcal{C}; \theta_{\mathcal{A}}) = \prod_{i=1}^{k} \prod_{j=1, j \neq i}^{k} P(r_{i,j} \mid c_i, c_j; \theta_{\mathcal{A}}). \tag{5}$$

---

[1]We implement an LLM-based RST parser $\mathcal{A}$ via prompting. Prompt is detailed in Appendix Figure 9.

[2]See Appendix Figure 10 for prompt and format details used in inter-chunk relation prediction.

Figure 2: The Discourse-RAG pipeline: Starting from passage retrieval (providing context), then intra-chunk RST tree parsing (capturing local discourse), inter-chunk rhetorical graph construction (modeling global discourse), rhetorical planning (structuring generation), and finally answer generation (producing the output).

As illustrated in the top-right panel of Figure 2, the graph $\mathcal{G}$ serves as a global discourse scaffold, allowing the generator to reason over cross-chunk connections.

**Rhetorically-Driven Generative Planning.** To move beyond the flat concatenation of retrieved evidence, we introduce a planning module that produces a rhetorically informed blueprint to guide the text generation. This is modeled through a mapping from the input query $q$, retrieved chunks $\mathcal{C}$ together with their RST trees $\mathcal{T}$, and the inter-chunk rhetorical graph $\mathcal{G}$ into a rhetorical plan $\mathcal{B}$:

$$(q, \mathcal{C}, \mathcal{T}, \mathcal{G}) \xrightarrow{\mathcal{A}} \mathcal{B}, \tag{6}$$

As illustrated in the center-bottom panel of Figure 2, the plan $\mathcal{B}$ is dynamically conditioned on the discourse structures and the query.[3] The plan outlines reasoning steps that involve selecting salient content, organizing argumentative flow, and prioritizing supporting evidence.

**RAG Generation with Rhetorical Guidance.** The final stage of generation[4] is conditioned on four inputs: (1) the original text chunks $\mathcal{C}$; (2) the intra-chunk RST trees $\mathcal{T}$; (3) the inter-chunk rhetorical graph $\mathcal{G}$; and (4) the rhetorical plan $\mathcal{B}$. The objective is:

$$y = \arg\max_{y'} P\big(y' \mid q, C, \mathcal{T}, \mathcal{G}, \mathcal{B}\big), \tag{7}$$

where $y'$ denotes a candidate output and $y$ refers to the final output that maximizes the conditional probability.

## 4 EXPERIMENTS

**Evaluation Datasets.** We evaluate our method on three benchmarks, namely Loong (Wang et al., 2024a), ASQA (Stelmakh et al., 2022), and SciNews (Liu et al., 2024). Loong dataset focuses on knowledge-intensive reasoning with Spotlight Locating (Spot.), Comparison (Comp.), Clustering (Clus.), and Chain of Reasoning (Chain.). These tasks are conducted under varying document lengths, where longer inputs increase evidence fragmentation and reasoning difficulty. ASQA involves long-form question answering and requires models to

---

[3]Appendix Figure 11 provides the prompt templates used in rhetorical planning.

[4]Appendix Figure 12 contains the generation prompt.

| Retrieval | Model | Spot. | | Comp. | | Clus. | | Chain. | | Overall | |
|---|---|---|---|---|---|---|---|---|---|---|---|
| | | LLM Score↑ | EM↑ | LLM Score↑ | EM↑ | LLM Score↑ | EM↑ | LLM Score↑ | EM↑ | LLM Score↑ | EM↑ |
| | | *Set 1 (10K–50K Tokens)* | | | | | | | | | |
| *Full Context* | Llama-3.1-8B-Instruct | 55.43 | 0.35 | 56.06 | 0.36 | 47.41 | 0.08 | 65.66 | 0.37 | 56.16 | 0.30 |
| | Llama-3.3-70B-Instruct | 58.82 | 0.44 | 61.33 | 0.35 | 48.15 | 0.11 | 70.31 | 0.37 | 59.54 | 0.32 |
| *Stradard RAG* | Llama-3.1-8B-Instruct | 62.61 | 0.32 | 60.61 | 0.26 | 53.61 | 0.08 | 58.76 | 0.32 | 60.08 | 0.25 |
| | Llama-3.3-70B-Instruct | 68.44 | 0.45 | 65.32 | 0.39 | 55.30 | 0.12 | 66.48 | 0.36 | 62.78 | 0.34 |
| *SOTA Results* | RQ-RAG* (Chan et al., 2024) | 72.31 | 0.54 | 48.16 | 0.05 | 47.44 | 0.07 | 58.96 | 0.25 | 53.51 | 0.17 |
| | GraphRAG* (Edge et al., 2024) | 31.67 | 0.00 | 27.60 | 0.00 | 40.71 | 0.14 | 54.29 | 0.43 | 40.82 | 0.18 |
| | StructRAG (Li et al., 2025b) | 74.53 | 0.47 | 75.58 | 0.47 | 65.13 | 0.23 | 67.84 | 0.34 | 69.43 | 0.35 |
| | Discourse-RAG (Llama-3.1-8B-Instruct) | 73.38 | 0.42 | 73.61 | 0.39 | 64.47 | 0.14 | 68.03 | 0.36 | 69.21 | 0.33 |
| | Discourse-RAG (Llama-3.3-70B-Instruct) | 76.62 | 0.45 | 75.66 | 0.46 | 65.38 | 0.19 | 68.29 | 0.38 | 71.01 | 0.37 |
| | | *Set 2 (50K–100K Tokens)* | | | | | | | | | |
| *Full Context* | Llama-3.1-8B-Instruct | 51.30 | 0.27 | 42.37 | 0.21 | 38.32 | 0.06 | 44.49 | 0.11 | 43.78 | 0.14 |
| | Llama-3.3-70B-Instruct | 55.27 | 0.34 | 47.93 | 0.26 | 40.05 | 0.08 | 50.08 | 0.10 | 48.24 | 0.17 |
| *Stradard RAG* | Llama-3.1-8B-Instruct | 57.02 | 0.25 | 45.42 | 0.19 | 44.21 | 0.05 | 50.42 | 0.15 | 49.12 | 0.16 |
| | Llama-3.3-70B-Instruct | 60.38 | 0.27 | 53.37 | 0.22 | 45.76 | 0.07 | 56.73 | 0.18 | 53.77 | 0.18 |
| *SOTA Results* | RQ-RAG (Chan et al., 2024) | 57.35 | 0.35 | 50.83 | 0.16 | 42.85 | 0.03 | 47.60 | 0.10 | 47.09 | 0.10 |
| | GraphRAG* (Edge et al., 2024) | 24.80 | 0.00 | 14.29 | 0.00 | 37.86 | 0.00 | 46.25 | 0.12 | 33.06 | 0.03 |
| | StructRAG* (Li et al., 2025b) | 68.00 | 0.41 | 63.71 | 0.36 | 61.40 | 0.17 | 54.70 | 0.19 | 60.95 | 0.24 |
| | Discourse-RAG (Llama-3.1-8B-Instruct) | 66.04 | 0.38 | 63.59 | 0.25 | 59.52 | 0.15 | 53.07 | 0.16 | 59.02 | 0.24 |
| | Discourse-RAG (Llama-3.3-70B-Instruct) | 69.93 | 0.40 | 64.36 | 0.36 | 61.68 | 0.18 | 58.25 | 0.21 | 63.62 | 0.29 |
| | | *Set 3 (100K–200K Tokens)* | | | | | | | | | |
| *Full Context* | Llama-3.1-8B-Instruct | 42.25 | 0.22 | 37.43 | 0.12 | 32.27 | 0.00 | 35.62 | 0.00 | 36.51 | 0.08 |
| | Llama-3.3-70B-Instruct | 47.31 | 0.31 | 41.11 | 0.14 | 35.64 | 0.01 | 49.78 | 0.01 | 42.27 | 0.11 |
| *Stradard RAG* | Llama-3.1-8B-Instruct | 49.22 | 0.21 | 40.24 | 0.03 | 36.04 | 0.00 | 49.05 | 0.00 | 43.42 | 0.06 |
| | Llama-3.3-70B-Instruct | 50.33 | 0.33 | 43.70 | 0.06 | 40.13 | 0.04 | 50.10 | 0.05 | 45.77 | 0.13 |
| *SOTA Results* | RQ-RAG* (Chan et al., 2024) | 50.50 | 0.13 | 44.62 | 0.00 | 36.98 | 0.00 | 36.79 | 0.07 | 40.93 | 0.05 |
| | GraphRAG* (Edge et al., 2024) | 15.83 | 0.00 | 27.40 | 0.00 | 42.50 | 0.00 | 43.33 | 0.17 | 33.28 | 0.04 |
| | StructRAG (Li et al., 2025b) | 68.62 | 0.44 | 57.74 | 0.35 | 58.27 | 0.10 | 49.73 | 0.13 | 57.92 | 0.21 |
| | Discourse-RAG (Llama-3.1-8B-Instruct) | 60.76 | 0.27 | 55.82 | 0.14 | 53.09 | 0.05 | 50.32 | 0.09 | 56.63 | 0.14 |
| | Discourse-RAG (Llama-3.3-70B-Instruct) | 66.39 | 0.39 | 57.83 | 0.28 | 58.87 | 0.08 | 52.19 | 0.16 | 58.88 | 0.23 |
| | | *Set 4 (200K–250K Tokens)* | | | | | | | | | |
| *Full Context* | Llama-3.1-8B-Instruct | 31.79 | 0.12 | 25.37 | 0.06 | 27.87 | 0.00 | 26.76 | 0.00 | 27.82 | 0.04 |
| | Llama-3.3-70B-Instruct | 36.76 | 0.21 | 32.22 | 0.07 | 30.69 | 0.00 | 30.17 | 0.00 | 32.21 | 0.05 |
| *Stradard RAG* | Llama-3.1-8B-Instruct | 40.01 | 0.11 | 31.90 | 0.00 | 32.33 | 0.00 | 29.92 | 0.00 | 33.52 | 0.02 |
| | Llama-3.3-70B-Instruct | 40.27 | 0.25 | 34.49 | 0.02 | 36.41 | 0.01 | 31.33 | 0.02 | 35.61 | 0.07 |
| *SOTA Results* | RQ-RAG* (Chan et al., 2024) | 29.17 | 0.08 | 40.36 | 0.00 | 26.92 | 0.00 | 34.69 | 0.00 | 31.91 | 0.01 |
| | GraphRAG* (Edge et al., 2024) | 17.50 | 0.00 | 26.67 | 0.00 | 20.91 | 0.00 | 33.67 | 0.33 | 23.47 | 0.05 |
| | StructRAG (Li et al., 2025b) | 56.87 | 0.19 | 55.62 | 0.25 | 56.59 | 0.00 | 35.71 | 0.05 | 51.42 | 0.10 |
| | Discourse-RAG (Llama-3.1-8B-Instruct) | 56.70 | 0.20 | 53.94 | 0.13 | 57.54 | 0.02 | 36.03 | 0.04 | 50.89 | 0.10 |
| | Discourse-RAG (Llama-3.3-70B-Instruct) | 67.77 | 0.26 | 55.82 | 0.19 | 57.39 | 0.03 | 36.10 | 0.07 | 54.63 | 0.13 |

Table 1: Loong benchmark results across four document-length settings. Our method (Discourse-RAG) is compared against zero-shot LLMs with full context, standard RAG, and prior SOTA. ⋆ indicates that the results are directly taken from Li et al. (2025b). We use **bold red** to indicate the best results and blue with underline to indicate the second-best results.

generate responses that are coherent and factually grounded. SciNews targets long-document summarization, where the objective is to rewrite scientific articles into accurate and accessible summaries for general audiences (Cachola et al., 2025). These datasets cover heterogeneous domains and provide a comprehensive evaluation of robustness and generalization. Dataset statistics are reported in Appendix Table 5.

**Evaluation Metrics.** To ensure consistency and fair comparison across, we follow the official evaluation protocols provided by each dataset's repository (Wang et al., 2024a; Stelmakh et al., 2022; Liu et al., 2024). For Loong dataset (Wang et al., 2024a; Li et al., 2025b), we report results using Exact Match (EM) and LLM-based scores. For ASQA (Stelmakh et al., 2022; Chang et al., 2025), the evaluation includes EM, ROUGE-L (RL) (Lin, 2004), and DR Score (Stelmakh et al., 2022). On SciNews, we evaluate with RL, BERTScore (Zhang et al., 2020), SARI (Xu et al., 2016), and SummaC (Laban et al., 2022). These metrics assess informativeness, fluency, and factual consistency. Detailed definitions are provided in Appendix C.

**Implementation Details.** Unless specified otherwise, we use Llama-3.1-8B-Instruct or Llama-3.3-70B-Instruct across all modules to instantiate and compare performance at different model scales (Grattafiori et al., 2024). For embedding and retrieval modules, we utilize Qwen3-Embedding-8B (Zhang et al., 2025c), using a chunk size of 256 tokens and Top-10 retrieval based on semantic similarity. Generation is performed using beam search with a beam width of 3. For Loong and ASQA, retrieval is conducted over the entire corpus, reflecting an open-domain setting. For SciNews, retrieval is restricted to the source document associated with each summary, reflecting a closed-domain setup.

**Selected Baselines.** We compare `Discourse-RAG` against three baseline settings: (1) zero-shot LLMs (`Llama-3.1-8B-Instruct` and `Llama-3.3-70B-Instruct`) with full input context. (2) standard RAG approach (Lewis et al., 2020), where relevant chunks are prepended to the query prior to inference.[5] and (3) previously published results from state-of-the-art RAG (if applicable) baselines on the same benchmarks.

## 5 RESULTS AND ANALYSIS

**General Results.** The experimental results are summarized in Table 1, Table 2, and Table 3, which correspond to the Loong, ASQA, and SciNews benchmarks, respectively. Across all benchmarks and evaluation metrics, `Discourse-RAG` consistently delivers stable and substantial improvements over the standard RAG baseline.

On the Loong benchmark, `Discourse-RAG` exhibits clear gains across varying document length settings. With `Llama-3.3-70B-Instruct` as backbone, our method achieves an LLM Score of 71.01 in Set 1, outperforming standard RAG by 8.23 points. The performance gap becomes more significant in Set 4, where `Discourse-RAG` scores 54.63 compared to 35.61 from standard RAG. When averaged across all four sets, our approach also surpasses the best prior reported training-based method `StructRAG`, thereby highlighting its robustness in long-context reasoning.

On ASQA, our method again yields consistent advantages. With `Llama-3.1-8B-Instruct`, EM, RL, and DR Score increase from 37.3/36.9/23.4 to 40.6/42.3/32.7, and with `Llama-3.3-70B-Instruct`, EM rises to 42.1 and DR to 33.0. Notably, our method outperforms `MAIN-RAG` (42.0 RL) and `Tree of Clarifications` (39.7 RL), achieving 42.4 RL score. On the SciNews summarization task, our approach exhibits strong generalization ability. Using `Llama-3.3-70B-Instruct`, `Discourse-RAG` obtains 21.12 RL score, 65.70 BERTScore, 44.39 SARI, and 69.49 SummaC, surpassing both standard RAG and the previous best system (Liu et al., 2024; 2025b).

| Model | EM↑ | RL↑ | DR Score↑ |
|---|---|---|---|
| Baselines with full context | | | |
| Llama-3.1-8B-Instruct | 20.1 | 30.6 | 16.3 |
| Llama-3.3-70B-Instruct | 22.7 | 32.9 | 16.8 |
| Baselines with standard RAG | | | |
| Llama-3.1-8B-Instruct | 37.3 | 36.9 | 23.4 |
| Llama-3.3-70B-Instruct | 38.2 | 37.2 | 24.1 |
| SOTA Results | | | |
| FLARE (Jiang et al., 2023) | 41.3 | 34.3 | 31.1 |
| Tree of Clarifications (Kim et al., 2023) | — | 39.7 | **36.6** |
| Open-RAG (Islam et al., 2024) | 36.3 | 38.1 | — |
| ConTReGen (Roy et al., 2024) | 41.2 | — | 30.3 |
| DualRAG (Cheng et al., 2025) | — | 31.7 | — |
| RAS (Jiang et al., 2025) | — | 39.1 | — |
| MAIN-RAG-Mistral-7B (Chang et al., 2025) | 35.7 | 36.2 | — |
| MAIN-RAG-Llama3-8B (Chang et al., 2025) | 39.2 | 42.0 | — |
| Ours | | | |
| Discourse-RAG (Llama-3.1-8B-Instruct) | 40.6 | 42.3 | 32.7 |
| Discourse-RAG (Llama-3.3-70B-Instruct) | **42.1** | **42.4** | 33.0 |

Table 2: Performance on the ASQA benchmark. `Discourse-RAG` consistently outperforms standard RAG baselines across all metrics. It also surpasses existing SOTA methods on most dimensions.

| Model | RL↑ | BERTScore↑ | SARI↑ | SummaC↑ |
|---|---|---|---|---|
| Baselines with full context | | | | |
| Llama-3.1-8B-Instruct | 15.33 | 59.27 | 35.43 | 48.31 |
| Llama-3.3-70B-Instruct | 17.19 | 61.03 | 37.65 | 54.73 |
| Baselines with standard RAG | | | | |
| Llama-3.1-8B-Instruct | 17.12 | 60.35 | 38.01 | 55.26 |
| Llama-3.3-70B-Instruct | 18.17 | 61.37 | 37.74 | 60.39 |
| SOTA Results | | | | |
| RSTformer Liu et al. (2024) | 20.12 | 62.80 | 41.56 | — |
| SingleTurnPlan Liang et al. (2024) | 19.68 | — | — | — |
| Plan-Input Liu et al. (2025b) | — | 65.32 | — | **72.40** |
| Ours | | | | |
| Discourse-RAG (Llama-3.1-8B-Instruct) | 19.26 | 63.49 | 40.27 | 63.37 |
| Discourse-RAG (Llama-3.3-70B-Instruct) | **21.12** | **65.70** | **44.39** | 69.49 |

Table 3: SciNews results. Our method (`Discourse-RAG`) improves over both zero-shot and RAG baselines, and often surpasses prior SOTA across multiple evaluation metrics.

**Ablation Studies.** We conduct ablation studies on the Loong benchmark, as summarized in Table 4, to assess the contribution of each component in `Discourse-RAG`. The removal of any single module, namely, the intra-chunk RST tree, the inter-chunk rhetorical graph, or the planning module, results in declines in

---

[5]All experiments are training-free and use only task instructions without in-context examples. Hyperparameters follow the settings described above.

| Method | Set 1 | | Set 2 | | Set 3 | | Set 4 | | Overall | |
|---|---|---|---|---|---|---|---|---|---|---|
| | LLM Score↑ | EM↑ | LLM Score↑ | EM↑ | LLM Score↑ | EM↑ | LLM Score↑ | EM↑ | LLM Score↑ | EM↑ |
| Discourse-RAG (full) | **71.01** | **0.37** | **63.62** | **0.29** | **58.88** | **0.23** | **54.63** | **0.13** | **62.12** | **0.26** |
| w/o RST tree | 65.47 | 0.35 | 58.42 | 0.22 | 54.92 | 0.17 | 47.67 | 0.09 | 56.24 | 0.21 |
| w/o Rhetorical graph | 67.81 | 0.35 | 58.89 | 0.25 | 54.07 | 0.17 | 48.19 | 0.11 | 57.11 | 0.22 |
| w/o Planning | 69.12 | 0.36 | 60.15 | 0.26 | 57.21 | 0.20 | 50.36 | **0.13** | 59.77 | 0.24 |
| Llama-3.3-70B-Instruct (standard RAG) | 62.78 | 0.34 | 53.77 | 0.18 | 45.77 | 0.13 | 35.61 | 0.07 | 49.33 | 0.17 |

Table 4: Ablation study of the three modules in `Discourse-RAG` with `Llama-3.3-70B-Instruct`. 'w/o RST tree' removes intra-chunk discourse modeling, 'w/o rhetorical graph' removes inter-chunk coherence modeling, and 'w/o planning' removes discourse-driven generative planning.

performance. The full model achieves an Overall LLM Score of 62.12, which falls to 56.24, 57.11, and 59.77 when the RST tree, rhetorical graph, and planner are removed, respectively. The Exact Match metric also decreases from 0.26 in the full setting to values ranging between 0.21 and 0.24 across the ablated variants.

Among the three components, the RST tree and rhetorical graph prove to be the most critical. In the long-document setting (Set 4), eliminating the RST tree leads to a decrease in LLM Score from 54.63 to 47.67. Similarly, removing the rhetorical graph reduces the score to 48.19, whereas excluding the planner causes a smaller drop to 50.36. These findings suggest that while all three modules contribute meaningfully, structural modeling within and across chunks plays a central role in aggregating information and maintaining discourse coherence in long-context generation.

**Impact of Retrieval Granularity and Noise Robustness.** To assess the robustness of `Discourse-RAG` under different retrieval conditions, we conduct a series of controlled experiments that manipulate three key variables: the chunk size of retrieved passages, the number of Top-$k$ passages, and the proportion of noisy (irrelevant) passages. All experiments are conducted on the Loong dataset using `Llama-3.3-70B-Instruct` as the unified generator. We maintain identical prompts and decoding configurations across all systems to ensure fair comparison. The evaluation includes two baseline methods, namely the full-context setting and the standard retrieval-augmented generation framework. Performance is reported using the aggregated LLM Score over four subsets, and the results are visualized in Figure 3.

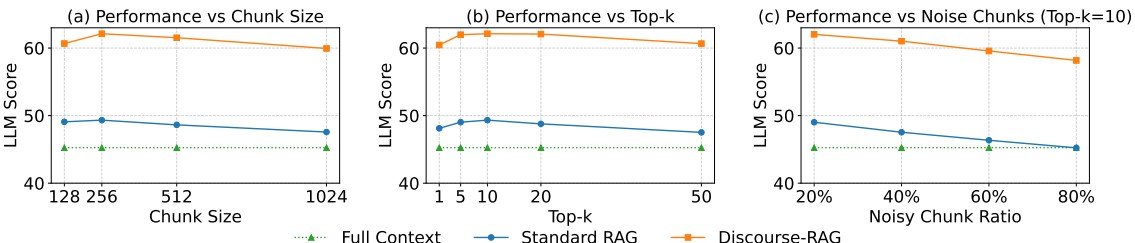

Figure 3: Retrieval stress test: performance under varying chunk size (a), Top-k value (b), and retrieval noise level (c), with identical prompts and decoding.

Panel (a) shows that standard RAG performs best at a moderate chunk size of 256 tokens (50.45) but suffers with larger chunks due to loss of structural coherence. In contrast, `Discourse-RAG` maintains stable performance across all chunk sizes, with scores ranging from 62.12 to 59.94, showing strong resilience to granularity shifts. Panel (b) examines that while standard RAG peaks at Top-10 and declines with larger $k$ due to accumulating noise, `Discourse-RAG` also performs best at Top-10 but remains robust up to Top-50, showing enhanced capacity to integrate and filter redundant information. Panel (c) evaluates noise robustness by replacing fractions of the Top-10 retrieved passages with unrelated content. In our experiments, we randomly replaced a certain proportion of the retrieved text chunks (*e.g.,* 20%, 40%) with irrelevant ones sampled at random from a pool of non-retrieved chunks. The standard RAG baseline exhibits a steep performance drop from 49.33 to 45.23 as noise increases, whereas `Discourse-RAG` retains a score of 58.17, highlighting the structural resilience of our method to retrieval errors.

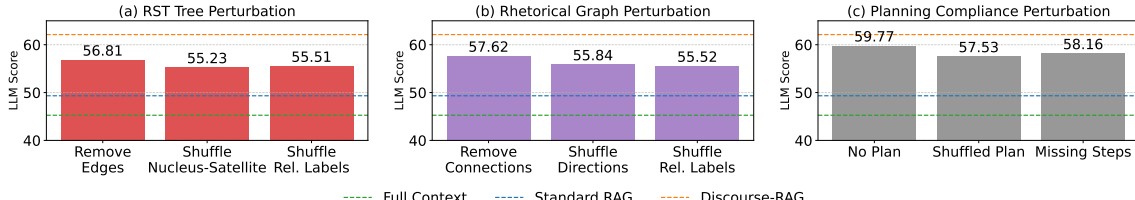

Figure 4: Effect of structural perturbations on performance. Panels (a), (b), and (c) correspond to intra-chunk RST trees, inter-chunk rhetorical graphs, and global rhetorical plans, respectively. Each perturbation involves randomly altering or removing the relevant elements.

**Impact of Structure Quality and Perturbation Causality.** To determine whether the performance gains of `Discourse-RAG` arise from the quality of structural modeling rather than the mere presence of structural cues, we conduct a set of controlled perturbation experiments targeting three core components of our framework. These include intra-chunk RST trees, inter-chunk rhetorical graphs, and global rhetorical plans. For each module, we introduce partial degradations by randomly selecting relation labels, edge directions, or planning steps, and either replacing or removing them. This design ensures that the perturbed structures still retain partial coherence, allowing us to assess how sensitive the model is to incomplete or noisy signals. All experiments are conducted with `Llama-3.3-70B-Instruct` under consistent retrieval and decoding conditions to maintain causal interpretability.

Figure 4 presents the results of the perturbation study. Panel (a) of Figure 4 shows that perturbing intra-chunk structures leads to consistent performance degradation. Randomly shuffling a portion of rhetorical relation labels reduces the LLM Score from 62.12 to 55.51. Randomly altering some nucleus–satellite roles lowers the score to 55.23, reflecting the model's sensitivity to rhetorical role assignments. Removing a randomly selected subtree connection decreases the score to 56.81, suggesting that structural completeness also contributes to generation quality. Panel (b) presents the effect of modifying rhetorical graphs. Randomly removing some graph connections between chunks reduces the score to 57.62. Randomly flipping the directions of a subset of edges yields 55.84, while replacing some discourse relation labels within the graph gives 55.52. These results suggest that both connection topology and relation semantics are integral to effective discourse-level modeling. Panel (c) analyzes the degradation of rhetorical plans. Omitting the plan altogether reduces performance to 59.77. Shuffling some of the step sequences causes a sharper decline to 57.53, while removing a subset of steps results in 58.16. These outcomes suggest that both the ordering and the completeness of the rhetorical plan are necessary for providing coherent structural guidance during generation.

Across all three dimensions, structural perturbations lead to measurable performance degradation, yet do not entirely eliminate the benefits conferred by structure-aware modeling. Even when exposed to corrupted or incomplete signals, `Discourse-RAG` consistently outperforms both the standard RAG baseline and the full-context setting. These results confirm that the observed improvements are not merely due to the inclusion of additional tokens, but instead arise from the model's capacity to leverage coherent and interpretable structural signals. Further discussion of LLM usage, limitations of our work, and qualitative case studies can be found in Appendix A, Appendix E, and Appendix F, respectively.

## 6 CONCLUSION

In this study, we tackle the absence of discourse structure modeling in existing RAG approaches by presenting `Discourse-RAG`. Grounded in Rhetorical Structure Theory, our approach constructs both local hierarchical and global discourse representations over retrieved evidence and leverages them to derive a high-level content plan that guides the reasoning process of the language model. Experimental results demonstrate that `Discourse-RAG` achieves significant gains across multiple knowledge-intensive QA and summarization tasks, surpassing previous state-of-the-art methods. Ablation studies further validate the complementary contributions of each structural component. Taken together, these findings highlight structured discourse modeling as a promising direction for advancing retrieval-augmented generation.

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

## A    THE USE OF LARGE LANGUAGE MODELS

In preparing this paper, we use GPT-5 as a writing assistant for language polishing, grammar correction, and stylistic refinement. The model is not involved in the research ideation, methodology design, experiments, or result interpretation. All technical content, analyses, and conclusions presented in this paper are fully conceived and validated by the authors. The authors take full responsibility for the content of the manuscript, including any parts generated with the assistance of GPT-5. In accordance with conference policy, we confirm that the LLM is not an author of this work and does not bear responsibility for its scientific claims.

## B    DETAILS OF DATASETS

Table 5 summarizes the key statistics of the Loong, ASQA, and SciNews datasets used in our experiments. The Loong dataset is a large-scale, cross-domain, and multi-task benchmark that covers long-text understanding, reasoning, and generation. It is specifically designed to evaluate models' ability in handling long contexts and performing comprehensive reasoning. The ASQA (Ambiguous Question Answering) dataset focuses on questions with multiple valid interpretations, providing explanatory responses that evaluate a model's capacity to resolve semantic ambiguity and produce interpretable answers. The SciNews dataset centers on the scientific news domain, spanning a wide range of scientific topics. It contains news articles with task-specific annotations and is intended to test models' capacity in long-context news understanding and summary generation.

| Dataset | Loong | | | | ASQA | SciNews |
|---|---|---|---|---|---|---|
| Spilt | Set1(10K-50K) | Set2(50K-100K) | Set3(100K-200K) | Set4(200K-250K) | Test | Test |
| Language | EN, ZH | EN, ZH | EN, ZH | EN, ZH | EN | EN |
| Test Instance | 323 | 564 | 481 | 232 | 1015 | 4188 |

Table 5: Summary statistics of the Loong, ASQA, and SciNews datasets used in our experiments.

## C    DETAILS OF EVALUATION METRICS

**For the Loong dataset.**    We report two evaluation metrics. The first is Exact Match (EM), which is a strict measure of the percentage of model predictions that exactly match any of the ground truth answers. It is a binary measure that assigns a score of one for a perfect match and zero otherwise. The second metric is the LLM Score (Wang et al., 2024a), ranging from 0 to 100. Following the protocol introduced by the dataset authors, we employ `GPT-4-turbo-2024-04-09` as an automated evaluator to rate the overall quality of generated responses. Unlike EM, which captures only factual correctness, the LLM Score provides a holistic evaluation by jointly considering comprehensiveness, clarity, and adherence to instructions, thereby offering a more integrated assessment across multiple dimensions of quality.

**For the ASQA dataset.**    We adopt the standard evaluation suite. The first is Exact Match (EM), defined as above. The second is ROUGE-L (Lin, 2004), a recall-oriented evaluation metric based on the Longest Common Subsequence (LCS). It measures the n-gram overlap between prediction and reference by identifying the longest sequence of words that occurs in both while preserving word order, thereby evaluating the coverage of key information. Given a predicted text $\hat{y}_i$ and a reference text $y_i$, let $LCS(\hat{y}_i, y_i)$ denote the length of their longest common subsequence. The ROUGE-L recall, precision, and F1 are defined as

$$R_L = \frac{LCS(\hat{y}_i, y_i)}{|y_i|}, \quad P_L = \frac{LCS(\hat{y}_i, y_i)}{|\hat{y}_i|}, \quad F_L = \frac{(1 + \beta^2) \cdot R_L \cdot P_L}{R_L + \beta^2 \cdot P_L}, \tag{8}$$

where $|y_i|$ and $|\hat{y}_i|$ are the lengths of the reference and predicted texts, respectively, and $\beta$ is set to one by default to balance recall and precision. In our experiments, we report ROUGE-L F1.

The third metric is the Disambiguation Recall (DR) Score (Stelmakh et al., 2022), which is specifically designed for ASQA to evaluate whether a prediction covers all possible disambiguated answers present in the reference set. While ROUGE-L cannot distinguish between two fluent but semantically divergent answers, the DR score explicitly evaluates coverage across multiple reference answers. A higher DR score indicates that the generated response captures a larger fraction of the possible interpretations of an ambiguous question.

Given multiple reference answers $\mathcal{Y}_i = \{y_i^{(1)}, y_i^{(2)}, \ldots, y_i^{(k_i)}\}$ for a query and a generated answer $\hat{y}_i$, the instance-level DR score is defined as:

$$\mathrm{DR}_i = \frac{1}{|\mathcal{Y}_i|} \sum_{j=1}^{|\mathcal{Y}_i|} \mathbf{1}\big[\hat{y}_i \text{ contains the information in } y_i^{(j)}\big], \tag{9}$$

where $\mathbf{1}[\cdot]$ is an indicator function equal to one if the predicted answer includes the content of a reference answer $y_i^{(j)}$, and zero otherwise. The overall DR score across $N$ queries is defined as:

$$\mathrm{DR} = \frac{1}{N} \sum_{i=1}^{N} \mathrm{DR}_i. \tag{10}$$

**For the SciNews dataset.** We focus on summarization quality using four metrics. The first is ROUGE-L (RL), as defined above. The second is BERTScore (Zhang et al., 2020), which computes token-level similarity between prediction and reference using contextual embeddings from pre-trained BERT models. Unlike n-gram–based metrics, BERTScore captures semantic similarity and often correlates more strongly with human judgment. The third is SARI (Xu et al., 2016), which assesses the quality of simplification by comparing system outputs against both the source text and the reference texts. SARI explicitly measures the precision and recall of words that are added, deleted, and kept. For a source sentence $s_i$, a prediction $\hat{y}_i$, and a set of reference simplifications $\mathcal{Y}_i = \{y_i^{(1)}, \ldots, y_i^{(k_i)}\}$, SARI is defined as:

$$\mathrm{SARI} = \frac{1}{3}\Big(\mathrm{Add}_{F_1} + \mathrm{Keep}_{F_1} + \mathrm{Del}_{F_1}\Big), \tag{11}$$

where $\mathrm{Add}_{F_1}$, $\mathrm{Keep}_{F_1}$, and $\mathrm{Del}_{F_1}$ denote the F1 scores for added, kept, and deleted n-grams relative to both the source and the reference sets. The fourth metric is SummaC (Laban et al., 2022), a model-based measure of factual consistency. SummaC can be used to determine whether a generated summary is entailed by its source document and detects unsupported or hallucinated content, which is essential for ensuring the reliability of generated text.

## D  DETAILS OF BASELINES

Here we describe the baselines used for comparison:

- **Standard RAG.** We implement the standard retrieval-augmented generation framework, where a retriever (`Qwen3-Embedding-8B`) retrieves relevant documents and a generator (`Llama-3.1-8B-Instruct` or `Llama-3.3-70B-Instruct`) produces the final answer conditioned on the retrieved context.
- **GraphRAG.** GraphRAG (Edge et al., 2024) augments retrieval with a graph-based knowledge representation by constructing a semantic knowledge graph from retrieved passages. It leverages community detection to capture global structures and integrates both local and global graph contexts into generation, enabling more accurate and globally coherent reasoning across documents.
- **RQ-RAG.** RQ-RAG (Chan et al., 2024) refines queries through explicit rewriting, decomposition, and disambiguation before retrieval. It trains LLMs end-to-end on a curated dataset with search-augmented supervision, enabling dynamic query refinement and improving both single-hop and multi-hop QA by learning to search only when needed.

- **FLARE.** Forward-Looking Active REtrieval augmented generation (FLARE) (Jiang et al., 2023) actively decides when and what to retrieve during generation by predicting upcoming sentences and using them as queries to fetch additional documents whenever low-confidence tokens appear.
- **Tree of Clarifications.** Tree of Clarifications (Kim et al., 2023) addresses ambiguous questions by recursively constructing a tree of disambiguated questions with retrieval-augmented few-shot prompting, pruning unhelpful branches through self-verification, and generating a long-form answer that covers all valid interpretations.
- **Open-RAG.** Open-RAG (Islam et al., 2024) enhances retrieval-augmented reasoning with open-source LLMs by transforming a dense model into a parameter-efficient sparse mixture-of-experts, combining contrastive learning against distractors with hybrid adaptive retrieval.
- **ConTReGen.** ConTReGen (Roy et al., 2024) employs a context-driven, tree-structured retrieval framework for open-domain long-form text generation. It performs top-down planning to recursively decompose a query into sub-questions for in-depth retrieval, followed by bottom-up synthesis to integrate information from leaf nodes to the root.
- **DualRAG.** DualRAG (Cheng et al., 2025) introduces a dual-process framework for multi-hop QA, consisting of Reasoning-augmented Querying (RaQ), which identifies knowledge gaps and formulates targeted queries, and progressive Knowledge Aggregation (pKA), which filters and structures retrieved information into a coherent knowledge outline. This closed-loop interaction enables dynamic adaptation to evolving knowledge demands and improves answer accuracy and coherence.
- **RAS.** Retrieval-And-Structuring (RAS) (Jiang et al., 2025) interleaves iterative retrieval planning with dynamic construction of query-specific knowledge graphs. It converts retrieved text into factual triples, incrementally builds a structured graph, and conditions generation on the evolving graph.
- **MAIN-RAG.** Multi-Agent Filtering RAG (MAIN-RAG) (Chang et al., 2025) is a training-free framework that employs three LLM agents to collaboratively filter and rank retrieved documents. It introduces an adaptive judge bar that dynamically adjusts relevance thresholds based on score distributions, effectively reducing noisy retrievals while preserving relevant information.
- **StructRAG.** StructRAG (Li et al., 2025b) introduces hybrid information structurization for knowledge-intensive reasoning. It employs a hybrid structure router to select the optimal structure type (e.g., table, graph, catalogue), a scattered knowledge structurizer to transform raw documents into structured knowledge, and a structured knowledge utilizer to decompose complex questions and infer accurate answers based on the structured representation.

# E LIMITATIONS AND FUTURE WORK

While `Discourse-RAG` demonstrates effectiveness across multiple benchmarks, we acknowledge several limitations that point toward promising avenues for future research.

First, our framework faces challenges in terms of computational efficiency. The training-free nature of `Discourse-RAG` comes at the cost of increased inference overhead. Specifically, the pipeline involves rhetorical structure parsing for each retrieved chunk, pairwise relation prediction across chunk pairs, global planning generation, and final answer generation, all of which rely on inference of LLMs. This leads to higher latency and computational cost per query compared to standard RAG methods (although the parsing of trees and graphs can be placed before retrieval). A key direction for future work lies in optimizing this pipeline, such as distilling a lightweight discourse parser or designing a unified multi-task model that jointly performs structural parsing and content generation in a single forward pass.

Second, the overall performance of our method is closely tied to the LLM's ability to generate high-quality rhetorical structures in zero-shot settings. Although our perturbation experiments suggest some robustness to noisy structures, we do not independently validate the intrinsic quality of the generated RST trees and

rhetorical graphs. This opens up risks of hallucinatory parsing and error propagation (i.e., inaccurate structural predictions may misguide planning and generation, resulting in factually inconsistent outputs). Future work should develop benchmarks for evaluating the reliability of LLM-generated structures. Additionally, moving beyond a fully training-free paradigm, one could fine-tune the discourse parser on high-quality corpora to improve structural fidelity and domain generalization.

Lastly, the current study leaves room for broader empirical coverage and evaluation scope. Our experiments focus primarily on long-document question answering and summarization. The generalizability of `Discourse-RAG` to other RAG applications, such as dialog-style QA, multilingual retrieval, and low-resource domains, remains to be tested. Moreover, exploring alternative discourse theories beyond RST, e.g., discourse graphs or neural coherence modeling, could help build more general and flexible structure-aware RAG frameworks.

## F  CASE STUDY

We present a qualitative case study to illustrate the effectiveness of `Discourse-RAG` compared to standard RAG, and the details of these two cases are shown in Figures 5 and 6.

## G  PROMPTS FOR DISCOURSE-RAG

Figures 9, 10, 11 and 12 present the prompts used in Discourse-RAG.

987
988
989
990
991
992
993
994
995
996
997
998
999
1000
1001
1002
1003
1004
1005
1006
1007
1008
1009
1010
1011
1012
1013
1014
1015
1016
1017
1018
1019
1020
1021
1022
1023
1024
1025
1026
1027
1028
1029
1030
1031
1032
1033

---

### Case Study 1

**Query.** *When did The Lion King debut on Broadway?*

**Retrieved Chunks.**

```
[Chunk 1] The Lion King is a stage musical ... It debuted on July 8, 1997, in
Minneapolis, Minnesota at the Orpheum Theatre ...
[Chunk 2] ... The musical premiered on Broadway at the New Amsterdam Theatre in
previews on October 15, 1997 ...
[Chunk 3] ... The official Broadway opening was on November 13, 1997, at the New
Amsterdam Theatre ...
[Chunk 4] ... On June 13, 2006, the Broadway production moved to the Minskoff
Theatre, where it is still running ...
```

**Standard RAG Output.**

> The Lion King debuted on Broadway in 1997. It was first performed in Minneapolis before moving to Broadway, where it became one of the most popular shows.

**Discourse-RAG Output.**

> The Lion King opened on Broadway more than once. It premiered in previews at the New Amsterdam Theatre on October 15, 1997, and officially opened on November 13, 1997. Later, on June 13, 2006, the production moved to the Minskoff Theatre, where it continues to run.

**Reference Answer**

> The Lion King opened on Broadway more than once. It premiered on Broadway at the New Amsterdam Theatre in previews on October 15, 1997, then officially opened on Broadway on November 13, 1997. On June 13, 2006, the Broadway production moved to the Minskoff Theatre, another Broadway theater, where it is still running after more than 9,000 performances.

**Remark.** Standard RAG produces a vague answer that misses key details about the Broadway timeline, providing only the year 1997. In contrast, `Discourse-RAG` integrates evidence across all relevant chunks, capturing both the preview and official opening dates as well as the later theater relocation, thereby delivering a more precise and comprehensive response.

Figure 5: Case study comparing standard RAG and `Discourse-RAG` on the query "When did The Lion King debut on Broadway?". Our method captures both the preview and official opening as well as the later relocation, while standard RAG gives only a vague year-based answer.

**Case Study 2**

**Query.** *Where did Italy win the World Cup in 2006?*

**Retrieved Chunks.**

```
[Chunk 1] The 2006 FIFA World Cup final was played on July 9, 2006, between Italy
and France ... after a 1-1 draw, Italy won 5-3 on penalties ...
[Chunk 2] ... The final was held at the Olympiastadion in Berlin, Germany, with
an attendance of 69,000 spectators ...
[Chunk 3] Italy claimed their fourth world title, having previously won in 1934,
1938, and 1982 ...
[Chunk 4] ... France had won their only previous World Cup in 1998, while Italy
had lost finals in 1970 and 1994 ...
[Chunk 5] The match was part of the France-Italy football rivalry, and marked
Italy's first world title in 24 years ...
```

**Standard RAG Output.**

> Italy won the 2006 World Cup in Munich, Germany, after a penalty shootout against France.

**Discourse-RAG Output.**

> Italy defeated France in the 2006 FIFA World Cup Final, played on July 9, 2006, at the Olympiastadion in Berlin, Germany. After a 1–1 draw, Italy prevailed 5–3 in the penalty shootout.

**Reference Answer.**

> The Italy national football team defeated France and was declared world champions in the 2006 World Cup Final, the last match of the 2006 FIFA World Cup, at the Olympiastadion stadium in Berlin, Germany, on July 9, 2006. Italy had won three previous finals in 1934, 1938, 1982 and lost in 1970 and 1994, while France had won their only previous final in 1998.

**Remark.** Standard RAG incorrectly states that the final took place in Munich, likely due to spurious retrieval or shallow aggregation of context. In contrast, Discourse-RAG integrates evidence across multiple chunks, correctly identifying the Olympiastadion in Berlin as the venue and providing richer historical context. This illustrates how explicit discourse modeling mitigates error propagation and enhances factual accuracy.

Figure 6: Case study comparing standard RAG and our proposed Discourse-RAG on the query "Where did Italy win the World Cup in 2006?". Our method correctly identifies the Olympiastadion in Berlin, while standard RAG produces a factual error.

---

**Relation Definitions in Intra-chunk RST Tree Construction**

**Relation Definitions:**
- ELABORATION: Satellite provides additional detail or information about the nucleus.
- EXPLANATION: Satellite explains or clarifies the nucleus content.
- EVIDENCE: Satellite provides evidence or proof for the nucleus claim.
- EXAMPLE: Satellite gives a specific example of the nucleus concept.
- CONTRAST: Satellite presents opposing or contrasting information.
- COMPARISON: Satellite compares two or more entities or concepts.
- CONCESSION: Satellite acknowledges opposing viewpoint while maintaining main claim.
- ANTITHESIS: Satellite presents directly opposite or contradictory information.
- CAUSE: Satellite describes the cause of an event or situation.
- RESULT: Satellite describes the result or consequence of an action.
- CONSEQUENCE: Satellite shows the outcome following from the nucleus.
- PURPOSE: Satellite explains the intended goal or purpose.
- CONDITION: Satellite specifies conditions under which something holds.
- TEMPORAL: Satellite indicates temporal relationship between events.
- SEQUENCE: Satellite shows sequential order of events or actions.
- BACKGROUND: Satellite provides background context or setting.
- CIRCUMSTANCE: Satellite describes circumstances surrounding an event.
- SUMMARY: Satellite summarizes or generalizes the nucleus content.
- RESTATEMENT: Satellite restates the nucleus in different words.
- EVALUATION: Satellite provides evaluation or assessment of the nucleus.
- INTERPRETATION: Satellite offers interpretation of the nucleus content.
- ATTRIBUTION: Satellite attributes information to a source.
- DEFINITION: Satellite defines a term or concept.
- CLASSIFICATION: Satellite classifies or categorizes information.

Figure 7: Relation Definitions for Intra-chunk RST Tree Construction.

---

**Relation Definitions in Inter-chunk Rhetorical Graph Construction**

**Relation Definitions:**
- SUPPORTS: Chunk provides support or evidence for another chunk.
- CONTRADICTS: Chunk contradicts or opposes another chunk.
- ELABORATES: Chunk elaborates on information in another chunk.
- EXEMPLIFIES: Chunk provides examples for another chunk's concepts.
- CAUSES: Chunk describes causes for events in another chunk.
- RESULTS_FROM: Chunk describes results from another chunk's events.
- ENABLES: Chunk describes what enables another chunk's situation.
- PREVENTS: Chunk describes what prevents another chunk's situation.
- PRECEDES: Chunk describes events that precede another chunk.
- FOLLOWS: Chunk describes events that follow another chunk.
- SIMULTANEOUS: Chunk describes simultaneous events with another chunk.
- BACKGROUND_FOR: Chunk provides background context for another chunk.
- GENERALIZES: Chunk provides general principles for another chunk's specifics.
- SPECIFIES: Chunk provides specific details for another chunk's generalizations.
- COMPARES_WITH: Chunk compares information with another chunk.
- CONTRASTS_WITH: Chunk contrasts information with another chunk.
- SUPPLEMENTS: Chunk supplements information in another chunk.
- REPLACES: Chunk replaces or updates information in another chunk.
- MOTIVATES: Chunk provides motivation for another chunk's content.
- JUSTIFIES: Chunk justifies claims or actions in another chunk.
- UNRELATED: Chunk has no meaningful rhetorical or semantic relation to another chunk.

Figure 8: Relation Definitions for Inter-chunk Rhetorical Graph Construction.

---

**Prompt for Intra-chunk RST Tree Construction**

You are an expert in Rhetorical Structure Theory (RST) analysis. Your task is to analyze the given text and construct a precise RST TREE.
**Critical instructions:**
1. RST tree is a HIERARCHICAL TREE structure (not a graph or network).
2. Each internal node has exactly two children: one NUCLEUS (core) and one SATELLITE (support).
3. NUCLEUS contains the main information; SATELLITE provides supporting content.
4. Relations describe how the SATELLITE relates to the NUCLEUS.
5. Think carefully and output ONLY ONE complete RST tree. Do not provide multiple analyses or revisions.
**Allowed RST relations:**
ELABORATION, EVIDENCE, EXAMPLE, CONTRAST, COMPARISON, CONCESSION, ANTITHESIS, CAUSE, RESULT, CONSEQUENCE, PURPOSE, CONDITION, TEMPORAL, SEQUENCE, BACKGROUND, CIRCUMSTANCE, SUMMARY, RESTATEMENT, EVALUATION, INTERPRETATION, ATTRIBUTION, DEFINITION, CLASSIFICATION
**Relation definitions:**
{Relation Definition}
**Step-by-step process:**
1. Segment text into meaningful text segments (clauses, sentences, or coherent units).
2. Determine the most important segment (this becomes the root nucleus).
3. For each other segment, decide: Is it NUCLEUS (core) or SATELLITE (support)?
4. Assign one relation from the allowed list.
5. Build the binary tree bottom-up.
**Required output format:**
SEGMENTS:
[1] <first segment>
[2] <second segment>
...
[N] <Nth segment>
RST ANALYSIS:
RELATION(segment$_i$, segment$_j$): {RELATION TYPE}
...
TREE_STRUCTURE:
ROOT[1-N]
├── NUCLEUS[X]  (N)
└── SATELLITE[Y]  (S): {RELATION TYPE}

**Validation rules:**
- Each segment must be complete and meaningful.
- Relations must be chosen from the allowed list.
- Mark (N) for nucleus, (S) for satellite.
- Output exactly ONE complete tree.
TEXT TO ANALYZE: {chunk$_i$}
Now analyze the given text following this exact format. Output ONLY ONE complete RST tree:

---

Figure 9: Prompt for Intra-chunk RST Tree Construction. The complete relation definitions are provided in Figure 7.

---

**Prompt for Pairwise Discourse Relation Inference**

You are an expert in discourse analysis. Your task is to determine the rhetorical relation between two given text chunks. Each call to this prompt considers only one chunk pair, and your goal is to assess whether there is a directed discourse relation from $\text{CHUNK}_i$ to $\text{CHUNK}_j$.

**Task objective:**
Analyze the discourse function of $\text{CHUNK}_i$ with respect to $\text{CHUNK}_j$, and decide whether there exists a meaningful rhetorical relation from $\text{CHUNK}_i$ to $\text{CHUNK}_j$. If so, identify and label the relation. Otherwise, return UNRELATED.

**Relation direction:**
Always assume the direction is from $\text{CHUNK}_i$ (source) to $\text{CHUNK}_j$ (target). The relation type should reflect how the source chunk contributes rhetorically to the target.

**Allowed relation types:**
SUPPORTS, CONTRADICTS, ELABORATES, EXEMPLIFIES, CAUSES, RESULTS_FROM, ENABLES, PREVENTS, PRECEDES, FOLLOWS, SIMULTANEOUS, BACKGROUND_FOR, GENERALIZES, SPECIFIES, COMPARES_WITH, CONTRASTS_WITH, SUPPLEMENTS, REPLACES, MOTIVATES, JUSTIFIES, UNRELATED

**Step-by-step process:**
1. Carefully read both $\text{CHUNK}_i$ and $\text{CHUNK}_j$.
2. Identify the main claim, fact, or event expressed in each chunk.
3. Ask: does $\text{CHUNK}_i$ serve any discourse function relative to $\text{CHUNK}_j$?
4. If a rhetorical link exists, name the relation type. If not, return UNRELATED.

**Required output format:**
$\text{CHUNK}_i$ -> $\text{CHUNK}_j$: {RELATION_TYPE}

**Validation rules:**
- Output exactly one line.
- Use only the allowed relation types.
- Relation direction must be from $\text{CHUNK}_i$ to $\text{CHUNK}_j$.
- Output UNRELATED if no meaningful relation is present.

TEXT TO ANALYZE:
$\text{CHUNK}_i$: [Insert first chunk here]
$\text{CHUNK}_j$: [Insert second chunk here]
Now analyze the rhetorical relation from $\text{CHUNK}_i$ to $\text{CHUNK}_j$ and output the result:

---

Figure 10: Prompt for pairwise discourse relation inference. The model is given two text chunks and must determine whether a directed rhetorical relation exists from the first to the second. This prompt is intended to be invoked once per chunk pair during graph construction. The complete relation definitions are provided in Figure 8.

---

**Prompt for Rhetorically-Driven Generative Planning**

You are an expert in discourse-aware text generation. Your task is to produce a RHETORICAL PLAN — a natural language paragraph that outlines how the final answer should be organized.
**Inputs:**
1. The user query.
2. Retrieved text chunks.
3. Intra-chunk RST trees, capturing local rhetorical hierarchies.
4. The inter-chunk rhetorical graph, modeling cross-passage discourse flow.
**Critical instructions:**
1. The plan must be written as a continuous paragraph in natural language.
2. The plan should describe the intended organization of the final answer.
3. The plan must be dynamically adapted to the given query and evidence; do not follow a fixed template.
4. Avoid reproducing the content of the chunks; only outline how they will be used.
5. Output exactly ONE complete rhetorical plan.
**Required output format:**
PLAN: <one paragraph in natural language that describes the planned organization of the answer>
TEXT TO ANALYZE: {query, chunks, RST trees, rhetorical graph}
Now generate one rhetorical plan that organizes the answer coherently:

Figure 11: Prompt for Rhetorically-Driven Generative Planning.

---

**Prompt for Rhetorical-Guided RAG Generation**

You are an expert in retrieval-augmented generation with discourse knowledge. Your task is to generate a coherent and faithful answer by leveraging the following inputs:
**Inputs:**
1. The user query.
2. Retrieved text chunks.
3. Intra-chunk RST trees, capturing local rhetorical hierarchies.
4. The inter-chunk rhetorical graph, modeling cross-passage discourse flow.
5. A rhetorical plan that outlines the intended argumentative organization.
**Critical instructions:**
1. The answer must directly address the user's query.
2. Integrate evidence from multiple chunks, guided by their RST trees and rhetorical graph.
3. Follow the rhetorical plan for structuring the answer.
4. Maintain factual accuracy, logical coherence, and rhetorical clarity.
5. Output a continuous answer in natural language. Do not output trees, graphs, or plans.
**Required output format:**
ANSWER: <a single coherent paragraph or multi-paragraph answer grounded in discourse structures>
**Validation requirements:**
- The answer must be faithful to the retrieved content.
- The answer must be logically organized and reflect discourse-level coherence.
- Avoid verbatim repetition of chunks; instead synthesize and integrate them.
- Output exactly one complete answer.
TEXT TO ANALYZE: {query, chunks, RST trees, rhetorical graph, rhetorical plan}
Now generate the answer in natural language:

Figure 12: Prompt for Rhetorical-Guided RAG Generation.

