# OpenReview forum: "Discourse-Aware Retrieval-Augmented Generation via Rhetorical Structure Modeling"
_ICLR.cc/2026/Conference — ICLR 2026 Conference Withdrawn Submission_

### Official Review · Reviewer_sbPP · 2025-10-30

**Soundness:** 3
**Presentation:** 2
**Contribution:** 3
**Rating:** 2
**Confidence:** 4

**Summary:**

The paper proposes Discourse-RAG, a retrieval augmented generation pipeline that makes the model explicitly use discourse structure. It first parses each retrieved chunk into an RST like tree and then links chunks with rhetorical relations to show support, elaboration, or conflict. A final planning stage guides generation using this structure. In evaluation it outperforms standard RAG and other structure aware baselines on long context QA, ambiguous QA, and scientific summarization.

**Strengths:**

- It uses one pipeline where chunk level discourse trees feed into a cross chunk graph, and both are used to guide generation.

- The method is tested on three tasks, with two Llama models, in both open and closed settings, and it beats 2025 RAG baselines, including on ASQA.

- The ablations, noise tests, and chunk size tests show that the method improves across different settings.

**Weaknesses:**

- While training-free, the pipeline requires multiple LLM calls per query (RST parsing per chunk, O(k²) pairwise relation inference, planning, generation). Cost grows quickly with larger k and the paper should discuss the latency and token counts.

- I noticed that all LLM benchmarks use Llama 3.x models. Since Qwen was already used for embeddings, why not include Qwen models in the main comparisons as well?

- Discourse quality is not validated. All trees and relations come from an LLM prompt rather than a parser with known accuracy. If the LLM segments poorly the whole pipeline can weaken.

- Relation set may be too big. The method uses many fine grained discourse labels but does not show which ones actually help. A smaller set might work the same.

- Evaluation scope is narrow. Results are mostly on English, long context, clean inputs. It is unclear how well this works on multilingual or noisy data.

**Questions:**

- Did you evaluate the accuracy of your LLM-generated RST trees against gold-standard annotations (e.g., on RST-DT)?

- Have you considered integrating neural discourse parsers or non-RST frameworks (e.g., entity grids, coherence models)?

- In cases where Discourse-RAG underperforms standard RAG (if any), what are the failure modes? Are they due to incorrect rhetorical parsing, poor planning, or something else?

- Beyond automatic metrics (ROUGE, LLM Score), was there any human assessment of coherence, faithfulness, or readability? LLM-based scoring can be biased.

---

> ### Author Response · Authors · 2025-11-20
> **Response to Reviewer sbPP (1/3)**
>
> We thank the reviewer for the insightful comments and helpful suggestions. Below, we provide point-by-point responses and clarifications.
>
> **Weakness 1**: We thank the reviewer for raising the issue of increased large model calls during inference under high *Top‑k* settings. This concern is addressed in our response to Reviewer **FdYo**, where we present a quantitative analysis on the Loong dataset. Using a consistent setup (same GPU, retriever, generator, chunk size, and decoding parameters), we varied *Top‑k* ∈ {10, 20, 30, 50} and measured both average token usage and total latency, including all structural modules. For convenience, we reproduce the experimental results below:
>
> | Top‑k | Standard RAG (Token Cost) | Discourse‑RAG (Token Cost) | Standard RAG (Latency) | Discourse‑RAG (Latency) |
> | ----- | ------------------------- | -------------------------- | ---------------------- | ----------------------- |
> | 10    | 3.6k                      | 20.9k                      | 7.7s                   | 21.5s                   |
> | 20    | 6.3k                      | 75.8k                      | 13.6s                  | 36.6s                   |
> | 30    | 8.1k                      | 217.7k                     | 21.4s                  | 52.6s                   |
> | 50    | 13.4k                     | 582.8k                     | 32.5s                  | 74.2s                   |
>
> We further note that the primary source of overhead is inter-chunk rhetorical graph construction. This cost can be reduced to a single model call by adopting a **list-level prediction** strategy:
>
> | Metric            | Standard RAG | List-level Discourse-RAG | Pairwise-level Discourse-RAG |
> | ----------------- | ------------ | ------------------------ | ---------------------------- |
> | LLM Score (Loong) | 49.33        | 59.67                    | 62.12                        |
> | EM (Loong)        | 0.17         | 0.24                     | 0.26                         |
>
> As discussed in the main text and Appendix E, this overhead reflects a design trade-off that prioritizes generation quality. As shown in Tables 1–3, our method yields consistent improvements across benchmarks, with over +10 LLM Score gains on long-document and multi-hop tasks. We believe this cost is justified by the benefits of discourse-guided generation, which lies at the core of our framework. For practical deployment, techniques such as dynamic Top‑k selection, structure caching, and using smaller models can help balance efficiency and quality. We refer the reviewer to our full response to **FdYo** for additional implementation details.
>
> ------
>
> **Weakness 2**: We thank the reviewer for pointing out the absence of Qwen models as backbone LLMs in our experiments. We appreciate the suggestion and offer the following clarification.
>
> During manuscript preparation, we indeed considered testing the compatibility of our method with the Qwen family. Specifically, we evaluated Qwen2.5‑72B‑Instruct on the Loong benchmark, comparing both standard RAG and Discourse-RAG configurations, alongside results from LLaMA‑3.3‑70B‑Instruct. The findings are summarized below:
>
> | Model Type                  | Avg. LLM Score | Avg. EM |
> | --------------------------- | -------------- | ------- |
> | Qwen2.5‑72B (Standard RAG)  | 43.45          | 0.16    |
> | Qwen2.5‑72B (Discourse‑RAG) | 58.57          | 0.23    |
> | LLaMA3‑70B (Standard RAG)   | 49.33          | 0.17    |
> | LLaMA3‑70B (Discourse‑RAG)  | 62.12          | 0.26    |
>
> These results demonstrate that discourse modeling consistently improves performance when used with Qwen2.5‑72B, confirming the method’s model-agnostic adaptability. That said, under the same settings, LLaMA‑3‑70B yielded stronger absolute performance. For clarity and conciseness, we prioritized reporting LLaMA‑3‑70B results in the main text. Nonetheless, we emphasize that our method is orthogonal to the choice of base LLM and its core contribution lies in the design of discourse-aware structural modeling, which can be integrated in a plug-and-play manner across different architectures.

---

> > ### Author Response · Authors · 2025-11-20
> > **Response to Reviewer sbPP (2/3)**
> >
> > **Weakness 3 and Question 1**: We thank the reviewer for raising concerns about the quality of discourse structures in our method. We understand the core issues to be: (1) the structures are generated by an LLM without supervision, and (2) potential structural errors may affect generation quality. These concerns are addressed in Section 5 through controlled perturbation experiments and further discussed in our response to Reviewer **R11v**.
> >
> > Briefly, we applied targeted perturbations to intra-chunk RST trees, inter-chunk rhetorical graphs, and rhetorical plans, including random swaps of nucleus–satellite roles, relation labels, edge directions, and removal of planning. As shown in Figure 4, these perturbations reduce the LLM Score (e.g., from 62.12 to 55.23 when altering nucleus–satellite roles), indicating that gains rely on the *semantic utility* of structure rather than its mere presence.
> >
> > We also evaluated our LLM-based parser (LLaMA‑3.3‑70B‑Instruct) on the RST-DT benchmark under zero-shot conditions, achieving F1 scores of 70.4 (Span), 63.1 (Nuclearity), and 55.6 (Relation). Taken together with the perturbation results, this suggests that Discourse-RAG is robust to moderate structural noise and consistently outperforms standard RAG.
> >
> > Finally, we emphasize that our structural modeling is *modular* and *model-agnostic*. Our aim is not to rely on a specific parser, but to evaluate whether discourse structure can improve RAG generation. We welcome future work that explores alternative structural modeling strategies within this framework.
> >
> > ------
> >
> > **Weakness 4**: We thank the reviewer for the question regarding the size and granularity of the discourse-relation label set. This is closely related to Reviewer **kqc4**’s **Question 2**, and we refer the reviewer to that response for a full discussion. A brief clarification is included here.
> >
> > The relation set used in Discourse‑RAG follows the standard RST framework and includes widely recognized semantic relations such as *Elaboration*, *Contrast*, *Cause*, and *Condition*. While these labels are relatively fine-grained, they are established and commonly adopted in mainstream discourse analysis, rather than ad hoc extensions (see [1–5]).
> >
> > To address whether this granularity is necessary, we compared full structural modeling to a shallow baseline using only discourse markers (e.g., *but*, *however*, *because*). This baseline yielded only a modest improvement of ~1 point (**49.33 → 50.41**), far below the gains achieved using full structure (**62.12**).
> >
> > [1] Can we obtain significant success in RST discourse parsing by using Large Language Models? (Maekawa et al., EACL 2024)
> >
> > [2] Multilingual Neural RST Discourse Parsing (Liu et al., COLING 2020)
> >
> > [3] Mann, William C., and Sandra A. Thompson. Rhetorical structure theory: A theory of text organization. No. ISIRS87190. Los Angeles: University of Southern California, Information Sciences Institute, 1987.
> >
> > [4] Stede, Manfred, Maite Taboada, and Debopam Das. "Annotation guidelines for rhetorical structure." Manuscript. University of Potsdam and Simon Fraser University (2017).
> >
> > [5] Liu, Dongqi, et al. "Explanatory Summarization with Discourse-Driven Planning." *Transactions of the Association for Computational Linguistics* 13 (2025): 1146-1170.
> >
> > ------
> >
> > **Weakness 5**: We thank the reviewer for highlighting the limitations regarding language and data coverage. We acknowledge that all evaluations are conducted on English datasets, and this is explicitly noted as a limitation in Appendix E. This issue is not an oversight but a known constraint we actively recognize.
> >
> > That said, Discourse-RAG is designed to be *modular* and *adaptable*. The current LLM-based RST parser operates in a zero-shot setting and may generalize to other languages through prompt adaptation or few-shot examples. The structural injection module is decoupled from the generator and can be replaced by language-specific tools (e.g., HIT-CDT for Chinese, DiZer for Portuguese). In low-resource or noisy environments, cross-lingual structure distillation could also be explored as a potential extension.

---

> ### Author Response · Authors · 2025-11-20
> **Response to Reviewer sbPP (3/3)**
>
> **Question 2**: We thank the reviewer for suggesting alternative structural frameworks. We agree that this is a valuable direction for future work. Our current choice of the RST framework is motivated by its ability to provide explicit hierarchical structure, nucleus–satellite distinctions, and relation types, which are the features that are well-suited for multi-evidence generation tasks.
>
> As discussed in our response to Reviewer **kqc4**, we also evaluated a baseline using shallow discourse markers (e.g., *but*, *because*). While this approach yielded modest improvements over structure-free baselines, it fell short of the gains achieved with full structural modeling. These results suggest that fine-grained discourse structures offer more effective guidance for generation.
>
> ------
>
> **Question 3**: We thank the reviewer for the question regarding whether Discourse-RAG underperforms in certain cases, and what the causes might be. We understand the concern and would like to clarify that such issues should be interpreted in the context of overall trends rather than isolated examples.
>
> Across all main experiments, including four input-length settings in Loong, as well as the ASQA and SciNews benchmarks, Discourse-RAG consistently outperforms standard RAG. This pattern is also confirmed by our three ablation studies and structural perturbation experiments. In rare borderline cases where Discourse-RAG performs slightly worse, our analysis suggests the cause is typically inaccurate or incomplete rhetorical parsing. Two pieces of evidence support this:
>
> 1. In long-form Loong settings, removing intra-chunk RST trees or inter-chunk rhetorical graphs causes larger performance drops than removing the planning module, indicating that structure quality has a more fundamental impact.
> 2. In structural perturbation experiments (Figure 4), corrupting nucleus–satellite roles, relation labels, or edge directions consistently leads to LLM Score degradation. In contrast, reordering plan steps has a milder effect, often mitigated by local evidence.
>
> In other words, when performance drops occur, they are generally due to local structural instability. Because Discourse-RAG treats discourse signals as interpretable structural priors, the planning process can amplify errors when parsing quality is poor. Nonetheless, the consistent gains across all three benchmarks and supporting studies demonstrate the robustness of structural modeling overall.
>
> ------
>
> **Question 4**: We thank the reviewer for raising the issue of evaluation dimensions. We agree that automatic metrics alone may not fully reflect generation quality. To address this, we adopt a multi-faceted evaluation approach. In addition to ROUGE and LLM Score, we report Exact Match (EM), a model-agnostic metric that measures alignment with human references. EM scores are reported for both Loong and ASQA in Tables 3 and 4, offering an unbiased signal of factual correctness.
>
> We also provide qualitative examples in Appendix F, covering multi-hop QA and temporal reasoning tasks. These examples illustrate how Discourse-RAG reduces factual errors (e.g., entity confusion, temporal inconsistencies) and integrates conflicting evidence more effectively through structure-aware generation.
>
> To further supplement automatic metrics, we now include human evaluation results on the SciNews dataset. Specifically, we randomly sampled 15 test instances and asked three graduate students with computer science backgrounds to rate outputs from both Standard RAG and Discourse-RAG. The evaluation followed the setup in [6], assessing four dimensions: *Relevance*, *Simplicity*, *Conciseness*, and *Faithfulness*.
>
> The results below show that Discourse-RAG consistently outperforms the baseline across all four criteria, with especially notable gains in *Faithfulness* and *Conciseness*. These results will be incorporated into the revised version of the paper.
>
> | System        | Relevant ↑ | Simple ↑ | Concise ↑ | Faithful ↑ |
> | ------------- | ---------- | -------- | --------- | ---------- |
> | Standard RAG  | 1.87       | 2.12     | 1.60      | 1.67       |
> | Discourse-RAG | 2.40       | 2.43     | 2.27      | 2.53       |
>
> [6] SciNews: From Scholarly Complexities to Public Narratives – a Dataset for Scientific News Report Generation (Liu et al., LREC-COLING 2024)
>
> We hope these clarifications have addressed your concerns. Thank you again for your constructive review.

---

> > ### Comment · Reviewer_sbPP · 2025-11-24
> >
> > Thank you for the responses, I will revised my scores.

---

> > > ### Author Response · Authors · 2025-11-24
> > >
> > > Thank you for your positive update and for raising the score. If there are any remaining concerns that you feel are not fully addressed, please let us know, and we would be very happy to provide further clarification or additional analyses.

---

### Official Review · Reviewer_kqc4 · 2025-10-31

**Soundness:** 3
**Presentation:** 3
**Contribution:** 3
**Rating:** 6
**Confidence:** 3

**Summary:**

This paper introduces Discourse-RAG, a novel training-free framework designed to address a key limitation in standard Retrieval-Augmented Generation (RAG): the tendency to treat retrieved documents as a flat, unstructured "bag of facts." This "flat structure" problem leads to intra-chunk structural blindness and inter-chunk coherence gaps, hindering the model's ability to synthesize evidence and reason.

**Strengths:**

S1. The paper identifies a clear and important limitation of standard RAG (its "flat structure") and proposes a novel, linguistically-grounded solution that directly addresses it.
S2. The method achieves state-of-the-art performance on multiple, diverse benchmarks (long-doc QA, ambiguous QA, summarization), demonstrating its effectiveness and generalization ability.
S3. The paper is clear, well-illustrated, and reproducible thanks to the detailed appendices.

**Weaknesses:**

W1. This method is computationally expensive. The proposed pipeline requires an enormous number of LLM inference calls per query. As described in the methodology, for a top-k retrieval, the framework k calls for intra-chunk RST tree construction, k * (k – 1) calls for inter-chunk rhetorical graph construction and 1 call for planning.
W2. The entire framework's success is predicated on the LLM's ability to function as a high-quality, zero-shot RST parser. This capability is assumed, not proven.

**Questions:**

Q1: Why did the authors not include an intrinsic evaluation of the RST parser against a gold-standard dataset? How can we be confident that the generated structures are faithful and not just plausible-sounding hallucinations that happen to guide the LLM?
Q2: Did the authors compare the full, complex RST parsing against simpler structural signals? For example, what is the performance if only explicit discourse markers (e.g., "however", "because", "in contrast") are used to build the inter-chunk graph, without any RST tree parsing?

---

> ### Author Response · Authors · 2025-11-20
> **Response to Reviewer kqc4**
>
> We sincerely thank the reviewer for the thoughtful feedback. We discuss each of your concerns in detail below.
>
> **Weakness 1**: We thank the reviewer for pointing out the computational overhead under high *Top‑k* settings. As noted in Appendix E, our method involves $k$ intra-chunk RST parses and $k(k–1)$ inter-chunk relation predictions. We further quantify this cost in our response to Reviewer **FdYo**, using end-to-end latency measurements. Intra-chunk structures can be precomputed during indexing and cached. Inter-chunk relations can be approximated via a single list-level pass, reducing complexity from $O(k^2)$ to $O(1)$. While list-level reasoning is less effective than pairwise prediction, it still offers substantial improvements over standard RAG.
>
> ------
>
> **Weakness 2 and Question 1**: We appreciate the reviewer’s concern regarding potential over-reliance on zero-shot LLM-based RST parsing. The core question, as we understand it, is whether Discourse-RAG depends critically on idealized structural quality. We address this both empirically and conceptually.
>
> First, Section 5 presents ablation studies on the three structural components: intra-chunk trees, inter-chunk graphs, and planning modules, under controlled perturbations. As shown in Figure 4, randomly corrupting nucleus–satellite roles, rhetorical labels, or subtree links reduces the LLM Score from 62.12 to 55.23, 55.51, and 56.81, respectively. Despite these drops, performance remains well above that of standard RAG (49.33), indicating that Discourse-RAG benefits from structural guidance without requiring perfect parsing.
>
> Second, to assess parser reliability, we evaluate the LLaMA‑3.3‑70B‑Instruct parser under a zero-shot setting on the RST-DT benchmark. It achieves a Span F1 of 70.4, Nuclearity F1 of 63.1, and Relation F1 of 55.6 (see our response to Reviewer **R11v**.
>
> Finally, the parser in our framework is modular and can be replaced with stronger alternatives, reinforcing the flexibility and adaptability of the approach.
>
> ------
>
> **Question 2**: We thank the reviewer for this question. To assess whether full RST parsing offers meaningful benefits, we compare Discourse-RAG with a variant that constructs shallow inter-chunk links using explicit discourse markers. This baseline does not apply EDU segmentation or capture hierarchical dependencies; instead, it forms surface-level relations between adjacent or semantically related chunks based on connective cues such as *however*, *but*, *although*, *in contrast*, *therefore*, *because*, *as a result*, and *meanwhile*. These markers are predicted by LLaMA‑3.3‑70B‑Instruct under a zero-shot setup.
>
> All systems are evaluated under identical retrieval and generation settings on the Loong benchmark:
>
> | Method              | LLM Score | Exact Match |
> | ------------------- | --------- | ----------- |
> | Standard RAG        | 49.33     | 0.17        |
> | + Discourse Markers | 50.41     | 0.20        |
> | Discourse-RAG       | 62.12     | 0.26        |
>
> The results indicate that marker-based shallow structures lead to only modest gains over standard RAG. However, these improvements are limited, as discourse markers capture only surface-level cues and cannot represent implicit rhetorical relations (e.g., *Concession*, *Condition*, *Elaboration*) or nucleus–satellite roles, which are essential for content selection. For tasks requiring cross-paragraph reasoning and long-range coherence, these shallow signals provide insufficient guidance. In contrast, full RST parsing and rhetorical graph modeling enable the system to capture deeper discourse structures and yield substantially higher performance under the same conditions.
>
> We believe these clarifications address your concerns, and we thank you again for your valuable input.

---

> ### Comment · Reviewer_kqc4 · 2025-11-25
>
> Thank you for your reply. I'll keep the score.

---

> > ### Author Response · Authors · 2025-11-25
> >
> > Thank you for your acknowledgement. Please feel free to let me know if any further clarification would be helpful.

---

### Official Review · Reviewer_R11v · 2025-11-01

**Soundness:** 2
**Presentation:** 2
**Contribution:** 2
**Rating:** 4
**Confidence:** 4

**Summary:**

The paper proposes Discourse-RAG, a retrieval-augmented generation framework that explicitly models intra- and inter-chunk rhetorical structures using Rhetorical Structure Theory (RST) and rhetorical planning to improve coherence and factual consistency in long-context reasoning. It demonstrates strong empirical results on multiple benchmarks (Loong, ASQA, and SciNews) across varying document lengths, outperforming several state-of-the-art RAG baselines. While the approach is somewhat heavy and empirically oriented, its clear performance gains and conceptual novelty justify acceptance, provided reproducibility and efficiency details are strengthened.

**Strengths:**

1. The idea of introducing rhetorical trees and inter-chunk discourse graphs into RAG is original and conceptually well-motivated, bridging discourse analysis and generative reasoning.
2. The method is tested on diverse, long-context benchmarks with detailed ablations and robustness studies (chunk size, Top-k, noise, and structure perturbations), giving credibility to the empirical claims.
3. Discourse-RAG outperforms strong baselines (StructRAG, MAIN-RAG, RQ-RAG) in both accuracy (LLM Score, EM) and factuality (SummaC, SARI), particularly on large-context and noisy retrieval scenarios.

**Weaknesses:**

1. Both intra- and inter-chunk discourse structures rely on LLM prompting for RST parsing, raising concerns about reproducibility, cost, and stability.
2. While results show improvements, the paper doesn’t deeply explore why rhetorical modeling helps or how structural cues propagate through the generator beyond surface correlations.
3. The multi-agent setup (parsing, graphing, planning) introduces significant preprocessing latency and complexity, which may limit real-time or large-scale deployment; no efficiency analysis is reported.

**Questions:**

1. Provide a quantitative evaluation of RST parsing accuracy and its impact on final performance (e.g., noise sensitivity to incorrect discourse trees).
2. Include a runtime and cost comparison versus other RAG systems (e.g., StructRAG, MAIN-RAG) to demonstrate practical feasibility.

---

> ### Author Response · Authors · 2025-11-20
> **Response to Reviewer R11v (1/2)**
>
> Thank you for your constructive feedback. We have carefully considered your comments and have addressed each point in our revised manuscript. Below, we provide our responses, with the aim of enhancing clarity and transparency.
>
> **Weakness 1**: We appreciate the reviewer’s attention to the RST parsing process. We clarify that structural modeling in our framework is a *task-agnostic integration*, allowing users to replace the current parser with alternatives such as fine-tuned or domain-specific parsers.
>
> For reproducibility, we use LLaMA-3.3-70B-Instruct to illustrate a zero-shot, task-transferable setup. Its invocation method, input/output format, and integration details are provided in Section 3 and Appendix G. Our anonymous code repository (linked in the abstract) includes prompt templates, output formats, error handling, and caching mechanisms. The associated computational cost is discussed in Appendix E and further quantified in our response to Reviewer **FdYo**, including token usage and latency under varying *Top‑k* values.
>
> Regarding stability, Section 5 of the original manuscript reports a three-part empirical analysis. Figure 3 shows consistent gains under varying chunk sizes, *Top‑k* values, and retrieval noise. Notably, Figure 3c demonstrates that even when 40% of retrieved passages are replaced with noise, Discourse‑RAG maintains a +7 LLM Score advantage over standard RAG. Figure 4 evaluates structural perturbations, including random relation replacement, edge reversal, and subtree deletion, and shows performance degradation. For instance, randomly replacing RST labels reduces the score from 62.1 to 55.5, still outperforming standard RAG (49.3), indicating robustness to parsing errors.
>
> ------
>
> **Weakness 2**: We thank the reviewer for highlighting the need to better explain the effectiveness of discourse modeling and how structural signals influence generation. We would like to clarify that this is empirically discussed in Section 5, where we discuss how structural priors guide information flow during generation.
>
> Our approach introduces discourse modeling at two levels. At the *encoding stage*, intra-chunk RST trees and inter-chunk rhetorical graphs impose structural constraints over retrieved content. At the *planning stage*, a rhetorical blueprint provides a high-level outline that governs logical order and argument flow prior to decoding. This design enables generation beyond flat concatenation, allowing for discourse-aware reasoning and organization.
>
> We further analyze attention and decoding behavior on the SciNews dataset:
>
> | Model Configuration | Inter-layer Attention Entropy ↓ | SummaC ↑ |
> | ------------------- | ------------------------------- | -------- |
> | Full Discourse-RAG  | 3.68                            | 69.5     |
> | – Intra-chunk RST   | 4.39                            | 66.1     |
> | – Inter-chunk Graph | 4.46                            | 67.3     |
> | Standard RAG        | 6.21                            | 60.4     |
>
> We observe that as structural guidance is reduced, attention entropy increases, indicating less focus on salient elements (e.g., nucleus spans, key nodes). This shift suggests that structural signals shape not only input organization but also attention patterns during decoding. Improved SummaC scores further reflect enhanced factual consistency.
>
> Together, these findings help explain the model’s robustness under stress conditions (e.g., varying *Top‑k*, chunk size, retrieval noise; see Figures 3–4), as discourse structures impose soft constraints on content ordering and causal flow, promoting stable generation under noisy inputs. We have clarified this point in the revised manuscript.
>
> ------
>
> **Weakness 3**: We thank the reviewer for raising concerns regarding the overhead introduced by structural modules. An efficiency analysis is provided in our response to reviewer **FdYo**. Intra-chunk RST parsing is *query-independent* and can be performed during offline index construction. Inter-chunk rhetorical graphs can be reduced via list-level prediction. The planning module is executed once before generation. At this stage, our primary objective is to evaluate the impact of structural modeling on RAG generation quality, rather than to optimize for minimal end-to-end cost.

---

> > ### Author Response · Authors · 2025-11-20
> > **Response to Reviewer R11v (2/2)**
> >
> > **Question 1**: We thank the reviewer for the suggestion regarding parsing accuracy. In response, we evaluated the LLM-based RST parser used in Discourse-RAG (LLaMA-3.3-70B-Instruct, zero-shot) on the RST-DT benchmark, following the evaluation protocol from [1]:
> >
> > | Model                                        | Setting      | Span F1 | Nuclearity F1 | Relation F1 |
> > | -------------------------------------------- | ------------ | ------- | ------------- | ----------- |
> > | Fine-tuned RST Parser (Maekawa et al., 2024) | Supervised   | 79.8    | 70.4          | 60.0        |
> > | Our LLM-RST Agent (zero-shot)                | Unsupervised | 70.4    | 63.1          | 55.6        |
> >
> > These results show that our zero-shot parser achieves reasonable accuracy, particularly in span and nuclearity prediction, approaching supervised baselines. The relation-level F1 also demonstrates moderate generalization to rhetorical semantics without task-specific tuning.
> >
> > Although parser development is not the primary focus of Discourse-RAG, our framework is *parser-agnostic*. The analysis in the original manuscript shows that better parsing accuracy correlates with downstream performance improvements, highlighting the benefit of structural modeling regardless of parser choice. This clarification has been incorporated into the revised manuscript.
> >
> > [1] Can we obtain significant success in RST discourse parsing by using Large Language Models? (Maekawa et al., EACL 2024)
> >
> > ------
> >
> > **Question 2**: We thank the reviewer for raising concerns about inference cost and deployment feasibility. We note that existing methods, such as StructRAG and MAIN-RAG, have not released their model checkpoints or inference code, which prevents direct reproduction or fair runtime comparison.
> >
> > Given this limitation, we have clarified in the main text that the primary source of computational overhead in our approach stems from the *structural parsing* component. A detailed breakdown is provided in our response to Reviewer **FdYo**, which we respectfully refer the reviewer to for further details.
> >
> > We hope our responses and revisions adequately address your concerns. Thank you again for your thoughtful feedback.

---

### Official Review · Reviewer_FdYo · 2025-11-11

**Soundness:** 3
**Presentation:** 3
**Contribution:** 3
**Rating:** 6
**Confidence:** 3

**Summary:**

The paper presents Discourse-RAG, a rag framework that explicitly models discourse structures. It works via a three-stage pipeline: 1) constructing intra-chunk RST trees to identify core vs. supporting information, 2) building inter-chunk rhetorical graphs to model relationships, 3) using a discourse-aware planning module to generate a blueprint for the final answer. Experiments on ASQA, Loong, and SciNews benchmarks show that Discourse-RAG achieves strong performance.

**Strengths:**

The paper is well written, and gets strong performance with good baselines.
The method can be plugged in any setup without any fine-tuning.
The components are ablated.

**Weaknesses:**

There is no analysis on how the method scales (in terms of cost (tokens) and latency) with higher top-k settings.

**Questions:**

What are the tradeoff with offline indexing and creation of the RST trees for the whole dataset?
How does the method scale in terms of latency, tokens with respect to higher top-k, different chunk sizes, bigger documents?

---

> ### Author Response · Authors · 2025-11-20
> **Response to Reviewer FdYo**
>
> Thank you very much for your feedback and the insights provided. We have carefully considered your comments and have revised our manuscript accordingly. Below, we detail our responses to each highlighted point:
>
> **Weakness 1:**
> We thank the reviewer for pointing out the issue of cost and latency scalability under higher Top‑k settings. We provide clarifications and supporting results below.
>
> First, Appendix E of our original manuscript (*"Limitations and Future Work"*) acknowledges the additional inference cost introduced by structural modeling. To further assess scalability, we conducted an end-to-end cost analysis on the Loong dataset under consistent conditions: identical GPU setup (16 × NVIDIA A100 80GB), retriever, generator, decoding parameters, and chunk size. Only Top‑k is varied in {10, 20, 30, 50}. We report both the average token cost (prompt + output) and the total latency (retrieval, structure parsing, planning, and generation):
>
> | Top‑k | Standard RAG (Token Cost) | Discourse‑RAG (Token Cost) | Standard RAG (Latency) | Discourse‑RAG (Latency) |
> | ----- | ------------------------- | -------------------------- | ---------------------- | ----------------------- |
> | 10    | 3.6k                      | 20.9k                      | 7.7s                   | 21.5s                   |
> | 20    | 6.3k                      | 75.8k                      | 13.6s                  | 36.6s                   |
> | 30    | 8.1k                      | 217.7k                     | 21.4s                  | 52.6s                   |
> | 50    | 13.4k                     | 582.8k                     | 32.5s                  | 74.2s                   |
>
> As Top‑k increases, the cost of Discourse‑RAG grows faster than that of standard RAG. This is primarily due to the construction of inter‑chunk rhetorical graphs, which require pairwise relation prediction over retrieved chunks. While the theoretical complexity is quadratic in $k$, the actual token usage is partially mitigated by LlamaIndex’s caching and parallelization.
>
> Several mitigations are possible. Intra‑chunk parsing is query-independent and can be performed offline. For inter‑chunk structures, pairwise prediction can be approximated by a **list-level** pass, reducing inference to a single step.
>
> To evaluate this approximation, we compare the original Discourse‑RAG with a list-level variant. Despite a slight performance drop, the list-level variant still outperforms standard RAG:
>
> | **Metric**        | **Standard RAG** | **List-level Discourse-RAG** | **Pairwise-level Discourse-RAG** |
> | ----------------- | ---------------- | ---------------------------- | -------------------------------- |
> | LLM Score (Loong) | 49.33            | 59.67                        | 62.12                            |
> | EM (Loong)        | 0.17             | 0.24                         | 0.26                             |
>
> These results confirm that even with reduced overhead, incorporating discourse structure remains beneficial. Ultimately, this overhead reflects a trade-off between **computation** and **performance**. As shown in *Tables 1–3* of the original paper, modeling discourse structure significantly improves the system’s ability to integrate evidence and generate coherent responses.
>
> ---
>
> **Question 1:**
> We thank the reviewer for raising the question regarding offline indexing and structural construction.
>
> As described in *Section 3* of our manuscript, our framework separates structural modules into three agents. Among them, intra-chunk RST trees depend solely on local content and are thus well-suited for offline processing. These structures can be precomputed during corpus preprocessing, serialized into lightweight formats (e.g., JSON, GraphML), and cached. This approach is especially effective for static or semi-static corpora. The primary trade-off is additional storage for the structural indices.
>
> For scalability, we refer to our previous response. In brief, while inter-chunk rhetorical graphs are constructed online, the number of nodes remains modest (typically Top‑10 to Top‑50). Moreover, as demonstrated above, *list-level reasoning* can serve as an effective approximation of pairwise relation prediction, further improving efficiency.
>
> We hope these clarifications satisfactorily address your concerns. We appreciate the opportunity to improve our work based on your feedback, and we remain open to further discussion.

---

### Author Response · Authors · 2025-11-27
**General Response**

Dear Reviewers,

Thank you for your review and comments on our manuscript.  We have carefully considered all comments and revised the manuscript accordingly. Below, we summarize the major changes and additions made in response to your suggestions:

1. **Computational Cost and Scalability** (FdYo, R11v, kqc4, sbPP):  We supplemented a latency and token cost analysis under varying Top-k settings on the Loong dataset. To reduce complexity, we presented a list-level approximation and offline intra-chunk RST parsing alternatives; experiments show that this approximation significantly reduces computational cost with slight performance degradation, while still outperforming standard RAG and SOTA baselines.

2. **RST Parsing Reliability and Structural Robustness** (R11v, kqc4, sbPP): We evaluated our zero-shot RST parser (*LLaMA-3.3-70B-Instruct*) on the RST-DT benchmark (Span/Nuclearity/Relation F1 scores of 70.4/63.1/55.6, respectively), showing that its accuracy is close to that of the supervised parser. Combined with structural perturbation studies from our original manuscript, we argue that Discourse-RAG benefits from better structure parsing results but remains robust to parsing noise.

3. **Effectiveness of Structural Modeling** (R11v, sbPP): Analysis on SciNews dataset reveals that removing structural signals increases attention entropy and reduces factual consistency during decoding. Together with ablations on Top-k, chunk size, and noise, we find that rhetorical structures influence information flow throughout selection, arrangement, planning, and generation.

4. **Shallow Discourse Baselines** (kqc4, sbPP): We compared Discourse-RAG with a variant using shallow discourse markers (e.g., *however*, *because*), which yielded only marginal gains over standard RAG. These findings underscore the importance of full RST structures in modeling implicit relations and content hierarchy.

5. **Backbone Model Selection** (sbPP): We reported additional results using *Qwen2.5-72B-Instruct* model, where Discourse-RAG continued to show consistent improvements. This confirms the framework's model-agnostic design and compatibility with different LLM families.

6. **Evaluation Dimensions and Human Assessment** (sbPP): In addition to automatic metrics (ROUGE, EM, SummaC), we included a human evaluation on the SciNews dataset covering relevance, conciseness, readability, and factuality. Discourse-RAG outperformed standard RAG across all dimensions, further supporting its effectiveness.

We are grateful for your time and effort invested in the review and discussion stages. We believe the revisions have improved the clarity, completeness, and rigor of our paper, and we hope these clarifications may help you more comprehensively evaluate the value of this work.

Sincerely,

The authors of Paper 3252

---

### Note · Authors · 2025-12-31

I have read and agree with the venue's withdrawal policy on behalf of myself and my co-authors.